# Single-molecule imaging and molecular dynamics simulations reveal early activation of the MET receptor in cells

Yunqing Li[1,8], Serena M. Arghittu [2,3,8], Marina S. Dietz [1], Gabriel J. Hella [2], Daniel Haße[4], Davide M. Ferraris [5], Petra Freund[1], Hans-Dieter Barth[1], Luisa Iamele[6], Hugo de Jonge[6], Hartmut H. Niemann [4], Roberto Covino[2,3,7] ✉ & Mike Heilemann [1,3] ✉

Embedding of cell-surface receptors into a membrane defines their dynamics but also complicates experimental characterization of their signaling complexes. The hepatocyte growth factor receptor MET is a receptor tyrosine kinase involved in cellular processes such as proliferation, migration, and survival. It is also targeted by the pathogen *Listeria monocytogenes*, whose invasion protein, internalin B (InlB), binds to MET, forming a signaling dimer that triggers pathogen internalization. Here we use an integrative structural biology approach, combining molecular dynamics simulations and single-molecule Förster resonance energy transfer (smFRET) in cells, to investigate the early stages of MET activation. Our simulations show that InlB binding stabilizes MET in a conformation that promotes dimer formation. smFRET reveals that the in situ dimer structure closely resembles one of two previously published crystal structures, though with key differences. This study refines our understanding of MET activation and provides a methodological framework for studying other plasma membrane receptors.

The formation of oligomers of single-pass transmembrane (TM) receptors activates numerous fundamental cellular events. Oligomerization is often induced by ligand-binding, and its stabilization involves weak interactions along the whole length of the receptor-ligand complex, including the receptor-bound ligand, the extracellular domain (ECD), the TM domain, as well as intracellular regions such as the juxtamembrane (JM) or kinase domain[1–4]. The membrane anchorage of receptors impacts their dynamics significantly. It increases the local concentration of receptors, reduces translational and rotational freedom, and pre-orients the receptor correctly for oligomerization. Other biomolecules defining the receptor's native environment can also contribute to the correct assembly of a receptor-ligand complex, e.g., lipids or glycans. To fully understand how a receptor forms biologically active oligomers, therefore, requires studying the receptor in its native, complex membrane environment. However, this severely complicates a structural and biochemical analysis.

In vitro approaches that study soluble fragments, like the ligand-bound receptor ECD, while very informative, may not be able to capture the contribution that weak interactions have to the complex formation of membrane-embedded proteins. On the other hand, only a few methods allow in situ analysis of membrane proteins in a native membrane environment. While cryo-electron tomography (cryo-ET)

[1]Institute of Physical and Theoretical Chemistry, Goethe-University Frankfurt, Max-von-Laue-Str. 7, Frankfurt am Main, Germany. [2]Frankfurt Institute for Advanced Studies, Ruth-Moufang-Str. 1, Frankfurt am Main, Germany. [3]IMPRS on Cellular Biophysics, Max-von-Laue-Str. 3, Frankfurt am Main, Germany. [4]Department of Chemistry, Bielefeld University, Universitaetsstr. 25, Bielefeld, Germany. [5]Department of Pharmaceutical Sciences, University of Piemonte Orientale, Largo Donegani 2, Novara, Italy. [6]Department of Molecular Medicine, University of Pavia, Immunology and General Pathology Section, Via Ferrata 9, Pavia, Italy. [7]Institute of Computer Science, Goethe-University Frankfurt, Robert-Mayer-Str. 11-15, Frankfurt am Main, Germany. [8]These authors contributed equally: Yunqing Li, Serena M. Arghittu. ✉e-mail: covino@fias.uni-frankfurt.de; heilemann@chemie.uni-frankfurt.de

holds great promise for the future[5], fluorescence microscopy has already provided important insights[6-9]. Cutting-edge microscopy methods that can address single proteins in the context of an intact cell ideally complement in vitro methods, especially if structural models exist that allow strategic site-specific fluorophore labeling. However, these methods usually lack the structural resolutions necessary to formulate precise mechanistic hypotheses.

Here, we overcome this challenge by leveraging an integrative structural biology approach combining computational structural modeling, molecular dynamics (MD) simulations, and single-molecule microscopy experiments to reveal the structural dynamics of membrane receptors in cells. We applied this approach to the receptor tyrosine kinase MET, which functions as a signaling protein on the plasma membrane and regulates cell proliferation, migration, and wound healing[10,11]. Dysfunction of MET is observed for a variety of diseases, such as cancer[12], diabetes[13], and autism[14]. In addition, the bacterial pathogen *Listeria monocytogenes* targets MET to initiate host cell invasion[15]. Signaling of MET is initiated by binding of the physiological ligand HGF[16,17], its natural isoform NK1[17,18], or the bacterial ligand InlB[15,19]. Upon ligand binding, two MET receptors and two ligands assemble into a 2:2 complex, which facilitates the transphosphorylation of the two MET proteins within the complex and downstream signaling[20]. However, for both the endogenous and the bacterial ligand, the in situ structures and the structural dynamics of the activation mechanism have not been resolved yet.

Using our integrative structural biology approach, we investigated the mechanism of InlB-mediated MET activation in situ. Simulations showed that the binding of InlB induces an extended conformation of the MET stalk that facilitates the formation of a complex mediated by ligand-ligand contacts. Single-molecule FRET in U-2 OS cells revealed the organization of the (MET:InlB)$_2$ complex in situ in the plasma membrane. We used this information to refine the structural model of the 2:2 complex and the dimer interface with MD simulations, resulting in a comprehensive picture of the early events of MET receptor activation by *L. monocytogenes*. Our approach is generally applicable to distinguish between conflicting structural models of ligand-receptor complexes or to scrutinize putative complex structures predicted by low-resolution experiments or computational tools such as AlphaFold[21] and RoseTTaFold[22].

## Results

### The binding of InlB promotes an extended conformation of the MET ectodomain
The heavily glycosylated MET ECD consists of six domains: the Semaphorin (Sema), the plexin-semaphorin-integrin (PSI), and four repeated immunoglobulin-like IPT1-IPT4 (Ig-like, plexins, transcription factors) domains. The intracellular part consists of the JM and the tyrosine kinase (TK) domain and is connected to the ectodomain by a single TM helix[10,23] (Fig. 1A). Both HGF and InlB bind to the Sema domain, and InlB additionally interacts with the IPT1 domain of the receptor[17,24,25].

We first explored how the binding of the invasion protein InlB affects the structural dynamics of the MET ectodomain. We modeled the upper ectodomain, comprising the Sema, PSI, and IPT1 domains (Fig. 1A), and ran atomistic MD simulations. To identify the consequences of InlB binding, we compared the dynamics of ectodomain fragments in isolation and in complex with InlB (Fig. 1B). Considering only a fragment of the ectodomain allowed us to run longer simulations and accumulate more sampling for a greater statistical significance. We chose a minimal version of InlB, InlB$_{321}$, which comprises a cap, a leucine-rich repeat (LRR), and an inter-repeat (IR) region, and which activates MET signaling[19].

The simulations revealed that the binding of InlB causes a large reduction in the flexibility of the upper ectodomain of MET. In the absence of InlB$_{321}$, IPT1 explores different orientations with respect to

the Sema domain (Fig. 1C and Supplementary Fig. 1A). In particular, PSI acts as a lever between the Sema and IPT1 domains, mediating the interactions between the two domains (Supplementary Fig. 1B). To quantify MET structural dynamics, we introduced the angle θ, defined as the angle formed between the Sema and IPT1 domains (Fig. 1C). In the isolated upper ectodomain, the angle value quickly decreased, corresponding to a structural closing of the IPT1 domain on the Sema (Fig. 1C, D). In the complex, instead, InlB prevents the Sema and IPT1 domains from closing onto each other. The angle describing the opening between the two domains converges to an average value of $θ_b = 135°$ (Fig. 1C, D).

Surprisingly, the conformation assumed by the upper ectodomain of MET in the complex with InlB is very similar to the one of MET in the complex with the endogenous ligand HGF (Fig. 1D, E). Despite the remarkably different binding modes of these two ligands, the angle formed by the Sema and IPT1 domains in both structures is approximately 135°. Moreover, the equilibrated model of the isolated MET upper ectodomain aligns with the crystal structure of the HGF beta-chain in complex with MET[25] (PDB 1SHY, see Supplementary Fig. 1C).

The MD simulations revealed that the binding of InlB controls the overall conformation of the MET ectodomain. In the absence of the ligand, in three independent MD simulations, the chain of the IPT domains slowly deviated from a linear arrangement, forming a very compact conformation of the ectodomain (Fig. 1F, H). In 2 out of 3 replicas, the Sema domain moved close to the terminal IPT4 domain and, therefore, close to the membrane. In contrast, all three MET:InlB complex replicas maintained a stable extended conformation (Fig. 1G, H).

To probe the conformation of the monomeric MET under the restraint imposed by the membrane bilayer, we simulated the entire MET and MET:InlB inserted in a membrane (Supplementary Fig. 2A). The large size of the system did not allow us to sample a sufficient time scale to reach statistical convergence (Supplementary Fig. 2B). Despite the limited sampling, our results are consistent with the behavior of monomeric MET in solution: the InlB-bound receptor exhibited a more extended conformation compared to the isolated receptor (Supplementary Fig. 2C). Interestingly, our simulations of MET:InlB on the membrane indicated that a kink between the IPT3 and IPT4 domains leads to a preferred tilt angle of the receptor stalk relative to the membrane plane (Supplementary Fig. 2C). We observed the same kink also in the corresponding simulations in solution (Fig. 1G).

We additionally generated a quasi-atomistic model[26] of the full MET ectodomain with and without bound InlB (Fig. 1I and Supplementary Fig. 3A). This slightly less detailed model enables us to simulate the dynamics of the MET:InlB complex at a reduced computational cost, allowing us to accumulate more statistical evidence. When bound, the InlB kept the receptor's stalk extended for up to 3 μs in 4 out of 5 replicas (Fig. 1I and Supplementary Fig. 3B, C) as opposed to the isolated MET, which collapsed within the first μs of simulation in both atomistic and quasi-atomistic models.

Overall, both the atomistic and the quasi-atomistic simulations showed that the structural constraints imposed by the binding of InlB on the upper ectodomain propagate non-locally along the whole chain of IPT domains. In the extended conformation of the complex in the membrane, InlB is exposed and preferably located at the same height from the membrane, compatible with the formation of a ligand-mediated (MET:InlB)$_2$ homodimer.

### In situ, FRET reports the relative orientation of InlB in (MET:InlB)$_2$
We used smFRET to reveal the orientation of two InlB$_{321}$ molecules within the dimeric (MET:InlB)$_2$ complex directly in cells. First, we generated two variants of InlB$_{321}$ carrying a single cysteine residue either at position 64 (K64C mutant) termed "H" (head) or at position 280 (K280C mutant) termed "T" (tail) (Fig. 2A). Using maleimide

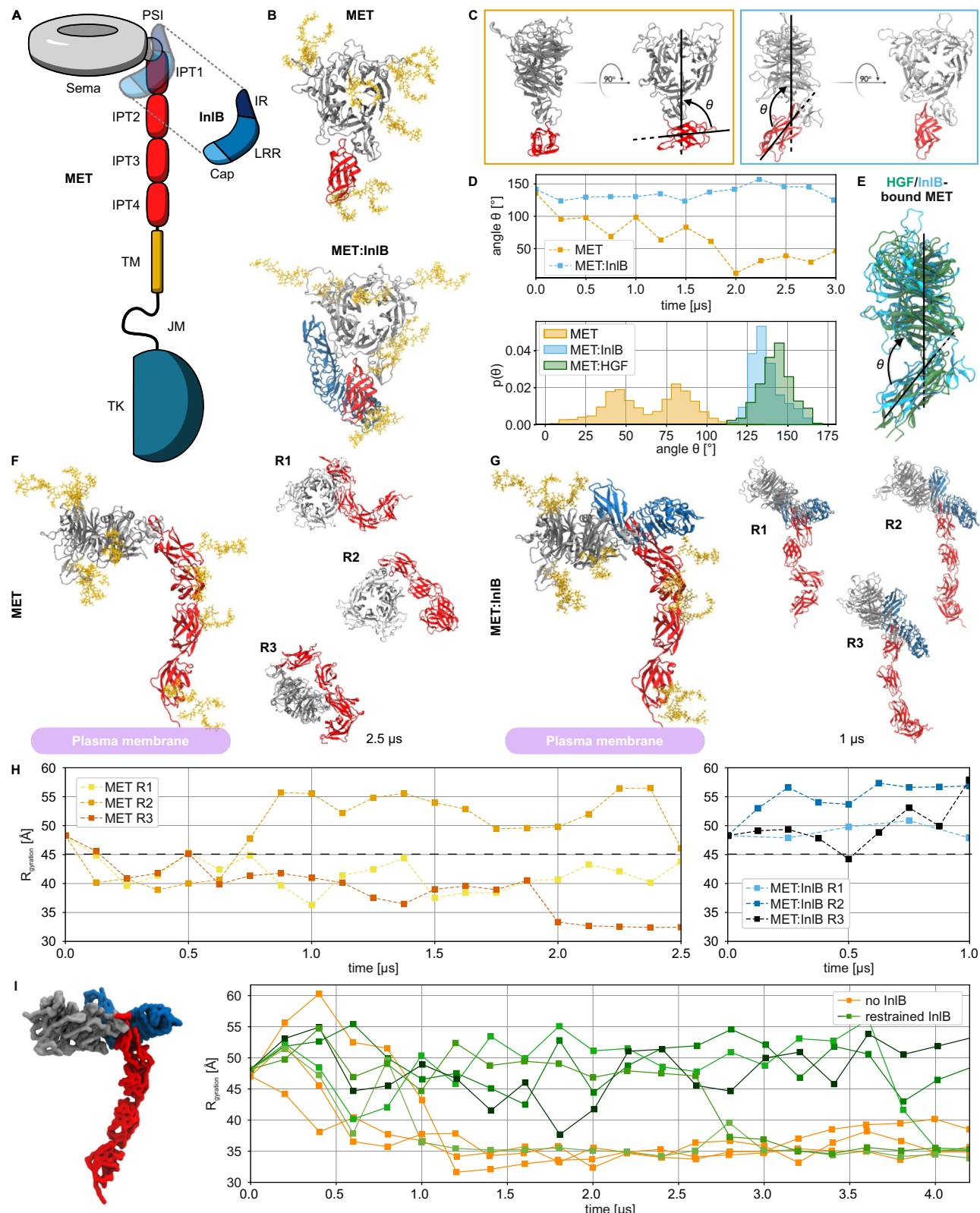

chemistry, we prepared fluorophore-labeled InlB$_{321}$ variants (Cy3B, ATTO 647N) in vitro and determined their degree of labeling (Supplementary Table 1). The activity of the fluorophore-labeled InlB$_{321}$ variants was determined from MET phosphorylation in U-2 OS cells using western blotting (Supplementary Fig. 4). The affinity of fluorophore-labeled InlB$_{321}$ was previously determined to be very similar to the unlabeled InlB$_{321}$[27].

Considering the proposed organizations of (MET:InlB)$_2$, two structural assemblies of the dimeric complex (MET:InlB)$_2$ are possible: a first one with the form I assembly (PDB 2UZX), and a second one with the form II assembly (PDB 2UZY)[24]. The fluorophore-labeled InlB constructs were designed to distinguish between these two forms by measuring three distances: H-H, T-H/H-T, and T-T (Fig. 2B, C). The expected donor-acceptor distances for two labeled InlB proteins in the

**Fig. 1 | Structural characterization of MET and MET:InlB$_{321}$ obtained with multiscale MD simulations. A** Schematic representation of the MET receptor bound to InlB$_{321}$. The ligand is represented transparently on the receptor structure. PSI: plexin-semaphorin-integrin, IPT: Ig-like, plexins, transcription factors, TM: transmembrane domain, JM: juxtamembrane, TK: tyrosine kinase, IR: inter-repeat region, LRR: leucine-rich repeat. **B** Renders of the N-glycosylated MET upper ectodomain system in isolation (top row, MET) and bound to InlB$_{321}$ (bottom row, MET:InlB$_{321}$). InlB is shown in blue; Sema and PSI in silver, the IPT1 in red, and the glycans in yellow; water and ions not shown for clarity). This representation is maintained for all renders except otherwise stated. **C** Side and front views of the conformations of MET (orange frame) and MET:InlB$_{321}$ (blue frame) at the end of the upper ectodomain simulations. **D** Time series of the θ angle of MET and MET:InlB models (top panel) and histograms of the θ angle calculated from simulations of the MET:InlB$_{321}$ model, the MET model, and the monomers in the MET

dimer in complex with its endogenous ligand HGF (based on PDB 7MO7) (bottom panel). **E** Render of one of the monomers involved in the MET:HGF dimer aligned to the MET:InlB$_{321}$ model. **F** Left: Render of the entire N-glycosylated MET ectodomain model in isolation. Right: Ectodomain configurations obtained by three replicas (R1–R3), each simulated for 2.5 μs. **G** Left: Render of the N-glycosylated MET entire ectodomain model bound to InlB$_{321}$. Right: Ectodomain configurations obtained by three replicas (R1-R3), each simulated for 1 μs. **H** Radius of gyration ($R_g$) computed on the Cα atoms of the replicas of the MET entire ectodomain model (yellow to red) and of the MET:InlB$_{321}$ entire ectodomain model (blue to black). The black dashed horizontal line at 45 Å indicates the threshold between extended ($R_g > 45$ Å) and collapsed conformations. **I** Left: render of the MET:InlB$_{321}$ model in quasi-atomistic resolution. Right: radius of gyration of the quasi-atomistic models of MET (orange) and MET:InlB$_{321}$ (green shades). Source data are provided as a Source Data file.

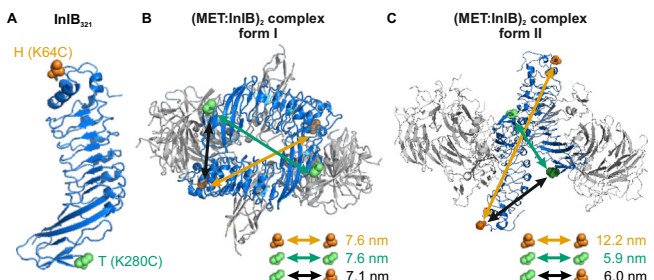

**Fig. 2 | InlB$_{321}$ site-specifically labeled variants in two possible MET:InlB dimer structures differing by the orientation of the MET:InlB monomers (MET in gray and InlB in blue). A** Two InlB variants, K64C (H variant, mutation site highlighted in orange) and K280C (T variant, mutation site highlighted in green), are labeled with donor and acceptor fluorophores for single-molecule FRET. The protein structure is adapted from PDB 1H6T. **B** Form I assembly of (MET$_{741}$:InlB$_{321}$)$_2$ dimer. Donor-acceptor distances between various combinations of two InlB variants were determined by AV simulations, yielding 7.6 nm (H-H), 7.6 nm (T-T), and 7.1 nm (T-H/H-T). The protein structure is adapted from PDB 2UZX. **C** Form II assembly of (MET$_{741}$:InlB$_{321}$)$_2$ dimer. Donor-acceptor distances between two InlB variants were determined by AV simulations, yielding 12.2 nm (H-H), 5.9 nm (T-T), and 6.0 nm (T-H/H-T). The protein structure is adapted from PDB 2UZY.

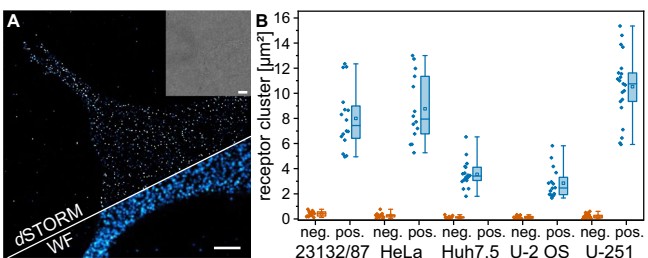

**Fig. 3 | Single-molecule super-resolution imaging of MET receptor densities at the plasma membrane of various cell lines. A** *d*STORM imaging of MET in U-2 OS cells. The super-resolution image (top), the widefield (WF) image (bottom), and the brightfield image (inset) are shown. The image is representative of four biological replicates. Scale bars 5 μm. **B** MET receptor cluster densities on the plasma membrane of different cell lines. The negative controls (orange) were obtained by incubating the cells with secondary antibodies without prior incubation with primary antibodies. The diamonds represent the receptor densities of single cells. The bounds of the boxes of the box plots display the 25th and 75th percentile, and the whiskers are the minima and maxima. In addition, the median (line) and the mean (square) are shown. Receptor densities were obtained from 16/13 (pos./neg.) (23132/87), 14/13 (HeLa), 19/12 (Huh7.5), 16/12 (U-2 OS), and 21/22 (U-251) cells from at least 3 independent experiments. Source data are provided as a Source Data file.

(MET:InlB)$_2$ complex were estimated by accessible volume (AV) simulations[28], yielding distances for the form I of 7.6 nm (H-H), 7.1 nm (T-H/H-T), and 7.6 nm (T-T), and for the form II of 12.2 nm (H-H), 6.0 nm (T-H/H-T), and 5.9 nm (T-T).

Next, we evaluated various adherent cell lines for their suitability to conduct smFRET microscopy of MET receptor complexes. This requires a MET surface density that is sufficiently low for spatial separation of single receptor assemblies with diffraction-limited microscopy. We measured the surface density of MET in various cell lines using *direct* stochastic optical reconstruction microscopy (*d*STORM)[29] (Fig. 3 and Supplementary Table 2). A first consideration was HeLa cells which are a standard cell line for studies of MET receptor[30–36]. However, the MET surface expression density in HeLa cells ranged between 6 and 14 clusters/μm² (Fig. 3B), which is too high for a spatial separation with diffraction-limited microscopy. In single-color imaging experiments, this limitation was bypassed by sub-stoichiometric labeling of MET with InlB$_{321}$[37]. However, smFRET experiments require both donor- and acceptor-labeled InlB$_{321}$, and sub-stoichiometric labeling would drastically reduce the probability of detecting donor-acceptor labeled (MET:InlB)$_2$ dimers (see also Supplementary Note 1). From the receptor density quantification (Fig. 3B and Supplementary Table 2), we selected U-2 OS as a cell model for smFRET imaging, because it showed the lowest density of MET on the plasma membrane with $2.8 \pm 1.2$ clusters/μm².

Next, we set up a smFRET experiment in fixed cells. We used a widefield microscope operated in total internal reflection (TIR) mode, in order to limit laser excitation close to the glass surface and thus the basal plasma membrane of cells[38] (Supplementary Fig. 5). In addition, we implemented alternating laser excitation (ALEX), where donor and acceptor fluorophores are excited in an alternating mode using two excitation wavelengths[38]. ALEX-FRET provides information on both the FRET efficiency (E) and the molecular stoichiometry (S) and enables "molecular sorting" in a two-dimensional E,S-histogram (Supplementary Fig. 5C). After U-2 OS cells were incubated for 15 min at 37 °C with 5 nM of both donor- and acceptor-labeled InlB, they were chemically fixed. Using the fluorophore-labeled H-/T-InlB$_{321}$ variants, samples for the three possible FRET pair combinations H-H, T-H/H-T, and T-T were prepared and measured with ALEX-FRET. FRET was detected for InlB$_{321}$ variant combinations T-T and T-H/H-T. Following accurate correction of experimental FRET data[39,40] (see "Methods"), we generated E,S-histograms and found a single population for both T-T and T-H/H-T (Fig. 4A). From the E,S-histograms, we extracted FRET efficiencies of $0.84 \pm 0.06$ (T-T) and $0.54 \pm 0.09$ (T-H/H-T), respectively (Fig. 4A, Supplementary Fig. 6, Supplementary Note 2 and Table 1). These FRET efficiency values correspond to distances of $4.8 \pm 0.3$ nm (T-T) and $6.2 \pm 0.4$ nm (T-H/H-T), respectively. Exemplary FRET time traces for single protein complexes show the expected acceptor photobleaching with a correlated rise in donor intensity (Fig. 4B). For cells that were labeled with H-/T-InlB$_{321}$, no FRET was detected (Supplementary Fig. 7). However, we detected colocalized fluorescence emission of Cy3B-H-InlB$_{321}$ and ATTO 647N-H-InlB$_{321}$ in single-molecule emission events (Fig. 4C and Supplementary Fig. 7B). Calculation of the FRET efficiencies and stoichiometries for single FRET pairs resulted in an E,S-histogram with a population close to a FRET efficiency of zero.

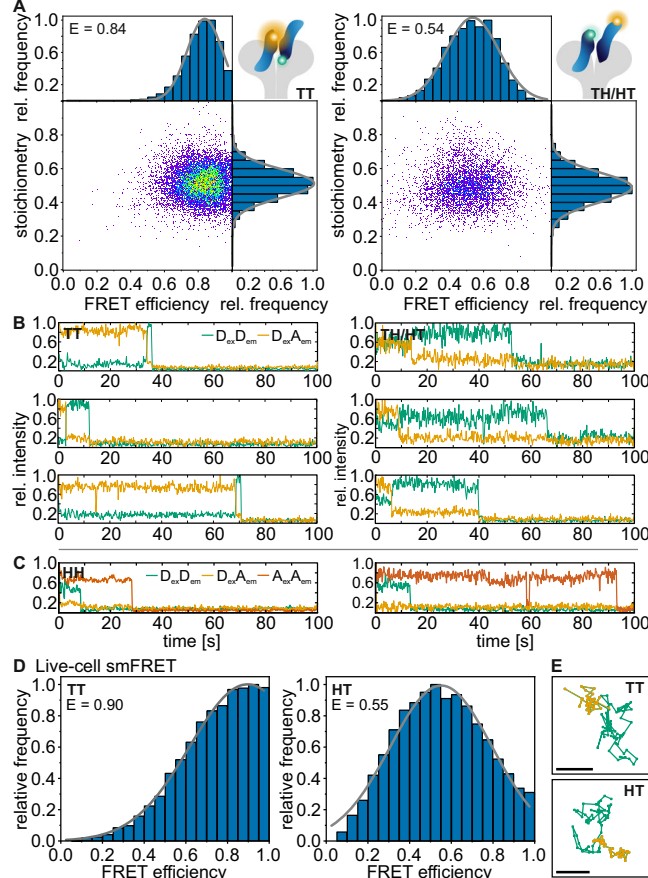

**Fig. 4 | Single-molecule FRET of (MET:InlB$_{321}$)$_2$ dimers in U-2 OS cells. A** smFRET with alternating laser excitation in fixed cells. Left: E,S-histogram for InlB T-Cy3B and T-ATTO 647N ($N = 113$ smFRET traces from 64 cells, 4 independent experiments); Right: E,S-histogram for InlB T-Cy3B and H-ATTO 647N and InlB H-Cy3B and T-ATTO 647N variants ($N = 49$ smFRET traces from 39 cells, 5 independent experiments). **B** Exemplary smFRET trajectories showing donor (green, D$_{ex}$D$_{em}$) and acceptor (orange, D$_{ex}$A$_{em}$) intensity traces (direct activation of acceptor not shown for clarity). **C** Exemplary single-molecule intensity traces extracted from colocalized spots showing the fluorescence signal of H-Cy3B and H-ATTO 647N variants showing donor (green, D$_{ex}$D$_{em}$), acceptor (orange, D$_{ex}$A$_{em}$), and direct excitation of the acceptor (red, A$_{ex}$A$_{em}$) fluorescence (see also Supplementary Fig. 7). Traces are normalized to 1. **D** Live-cell smFRET of (MET:InlB)$_2$ dimers. FRET efficiency histograms of Cy3B-T-InlB$_{321}$ and ATTO 647N-T-InlB$_{321}$ (left, $N = 564$ smFRET traces from 27 cells, 5 independent experiments) and for Cy3B-H-InlB$_{321}$ and ATTO 647N-T-InlB$_{321}$ variants (right, $N = 757$ smFRET traces from 24 cells, 3 independent experiments). **E** Exemplary trajectories of single FRET pairs of the T-T and H-T combination. The donor trajectory is shown in green, while the acceptor trajectory (acceptor emission upon donor excitation, i.e., FRET signal) is shown in orange and simultaneously represents the colocalized movement of the donor and acceptor. Scale bars 500 nm. Source data are provided as a Source Data file.

## Table 1 | Predicted and experimentally determined FRET efficiencies for different combinations of InlB$_{321}$ variants

|  | AV predicted FRET efficiency |  | Experimental FRET efficiency |  |
|---|---|---|---|---|
|  | Form I | Form II | Fixed cells | Live cells |
| T-T | 0.258 | 0.620 | 0.84 ± 0.06 | 0.90 ± 0.05 |
| T-H/H-T | 0.317 | 0.567 | 0.54 ± 0.09 | 0.55 ± 0.09 |
| H-H | 0.248 | 0.018 | 0.03 ± 0.06 | _(a) |

[(a)]No FRET signal was detectable for H-H labeled MET dimers because the FRET efficiency was so low that no anti-correlated behavior of donor and acceptor emission upon donor excitation could be observed.
Errors are given according to Agam et al. [55]

## Table 2 | MD-predicted distances for the three FRET dye pairs for each replica (R1, R2, R3) in comparison to the experimental values

| Dye pair[(a,b)] | R1 | R2 | R3 | Experiment |  |
|---|---|---|---|---|---|
|  |  |  |  | Fixed cell | Live cell |
| T-T | 3.6 ± 0.1 | 4.1 ± 0.1 | 4.2 ± 0.2 | 4.8 ± 0.3 | 4.4 ± 0.4 |
| T-H/H-T | 5.9 ± 0.1 | 5.8 ± 0.1 | 6.3 ± 0.1 | 6.2 ± 0.4 | 6.1 ± 0.4 (H-T) |
| H-H | 10.2 ± 0.1 | 9.8 ± 0.1 | 9.7 ± 0.1 | 11.4 ± 4.2 | _(c) |

[(a)]The errors on the predicted distances were computed by summing the variances describing the statistical contribution and the experimental uncertainty on the R$_0$ value and taking the square root of the resulting variance. The statistical uncertainty corresponds to a 66% confidence interval estimated by bootstrapping the distance distributions. We resampled the FRET efficiency 1000 times, computed using FRETpredict[45] with repetition, and calculated the mean of each sample. We converted the 1000 bootstrapped means to distances (using the experimental R$_0$ value for the FRET pair), and computed the standard deviation. For the T-H/H-T distance, we first converted the MD-calculated efficiency into distance prior to bootstrapping. We then applied bootstrapping to the merged distance series and computed the standard deviation. Our estimate of the uncertainty arising from the MD simulations does not include potential forcefield inaccuracies and should be considered a lower-bound estimate.
[(b)]The experimental FRET distance uncertainty is calculated according to Hellenkamp et al. [56].
[(c)]Live-cell FRET was measured using the smFRET-RAP approach. No FRET signal was detected for H-H labeled MET dimers in smFRET measurements in living cells.
All values are given in nm.

This population is clearly separated from the donor-only population by their different stoichiometries (Supplementary Fig. 7C). However, it should be mentioned that the FRET efficiency is so low that accurate distance calculation is not possible (Tables 1 and 2).

In order to evaluate the specificity of InlB targeting MET receptors, we used a single chain Fv (scFv) fragment of a previously published antibody (107_A07) that binds an epitope on MET IPT1 overlapping with the InlB binding site[41]. Thus, pre-incubating cells with the 107_A07 scFv will block InlB binding to MET. Our experiments showed that the scFv fragment efficiently blocked InlB binding, and we did not see background fluorescence (Supplementary Fig. 8). As additional control to prove the existence of InlB-mediated MET

dimers, we performed single-molecule photobleaching experiments. We labeled MET in U-2 OS cells with a single InlB variant, Cy3B-H-InlB$_{321}$ or Cy3B-T-InlB$_{321}$, and found two-step photobleaching (Supplementary Fig. 9).

To exclude deviations in FRET efficiency that might originate from the chemical fixation of cells, we performed smFRET experiments in living U-2 OS cells. We treated living cells with either a combination of Cy3B-T-InlB$_{321}$ and ATTO 647N-T-InlB$_{321}$ (T-T), Cy3B-H-InlB$_{321}$ and ATTO 647N-T-InlB$_{321}$ (H-T), or Cy3B-H-InlB$_{321}$ and ATTO 647N-H-InlB$_{321}$ (H-H), and measured FRET efficiencies of single FRET pairs using the smFRET recovery after photobleaching (smFRET-RAP) method[42] to reduce overlap of single-molecule signals and background (Supplementary Fig. 10). The smFRET analysis yielded single populations with FRET efficiencies of 0.90 ± 0.05 (T-T) and 0.55 ± 0.09 (H-T) (Figure 4DE), and as such similar values as obtained from fixed cell smFRET experiments (Fig. 4A). The FRET efficiency values correspond to distances of 4.4 ± 0.4 nm (T-T) and 6.1 ± 0.4 nm (H-T). For the H-H pair, no FRET signal was observed.

Beyond informing on distances, smFRET can report on structural flexibility. This information can be extracted from the width of the FRET efficiency distribution. We employed photon distribution analysis (PDA), which predicts the theoretical FRET distribution considering the setup-dependent shot noise and background[43,44]. PDA yields histograms for the ratio between donor emission and donor-excited acceptor emission that can be compared to experimental data. We found that the PDA histograms are largely identical to

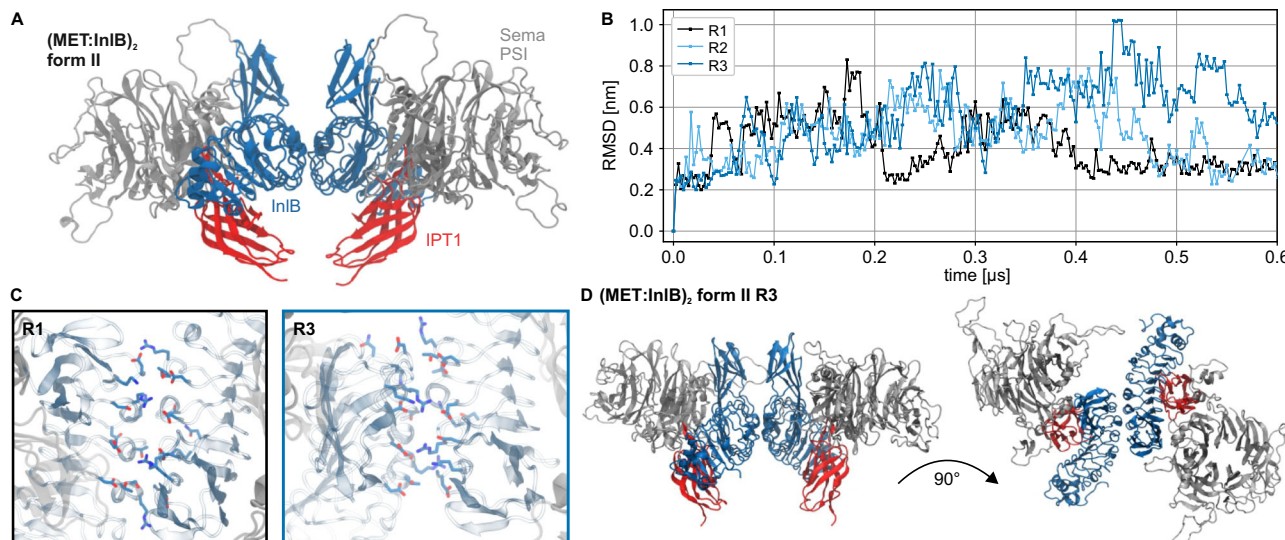

**Fig. 5 | Molecular dynamics simulation of the (MET:InlB₃₂₁)₂ dimer. A** Renders of the initial form II (MET:InlB₃₂₁)₂ complex model (Sema and PSI domain of MET in silver cartoon and IPT1 domain in red cartoon; InlB in blue cartoon. Water, ions, and glycans not shown for clarity). **B** RMSD time series of the form II (MET:InlB₃₂₁)₂ complex model replicas calculated with respect to the first frame. **C** Representative assemblies of the two different dimer interfaces explored during the simulations (InlB₃₂₁ in cyan cartoon, MET in silver cartoon, positive side chains in blue, negative side chains in red). Replica 1 explored a broader dimer interface (black frame), while Replica 3 explored a more compact one (blue frame; water, ions, and glycans not shown for clarity). **D** Render of the proposed antisymmetric dimer structure (explored by R3, compact dimer interface) showing top view (top panel) and side view (bottom panel) (water, ions, and glycans not shown for clarity). Source data are provided as a Source Data file.

experimental smFRET data in fixed cells (Supplementary Fig. 11), indicating very little structural flexibility in the MET dimer.

A comparison of the smFRET-derived FRET efficiencies to the AV predicted values for the respective fluorophore-labeled InlB variants (Table 1) and the finding that the H-H combination did not yield a FRET signal (Fig. 4C and Supplementary Fig. 7) suggested that the (MET:InlB)₂ dimer favors a form II assembly in cells. However, the FRET-derived distances are not in quantitative agreement with the values predicted from the crystal models (Fig. 2B, C). While the experimental result for the T-H/H-T distance (6.2 nm) is close to the predicted value (6.0 nm), the experimental result for the T-T distance (4.8 nm) is considerably shorter than the predicted value (5.9 nm). This discrepancy is larger than what is expected from the accuracy of AV simulations and motivated us to investigate the structure of (MET:InlB₃₂₁)₂ with MD simulations.

### MD simulations of the (MET:InlB)₂ dimer quantitatively explain the experimental FRET data

We performed atomistic MD simulations of the form II (MET:InlB)₂ dimer model. We started from the proposed form II structure (PDB 2UZY), containing two copies of the upper ectodomain in complex with InlB (Fig. 5A). In this model, back-to-back contacts between the two InlBs constitute the dimer interface. We then ran three independent replicas each for 600 ns to assess statistical variability. The dimer remained associated in all replicas and sampled only local rearrangements. One replica (R1) remained the closest to the initial starting structures, whereas the other two (R2 and R3) rearranged in a more significant way (Fig. 5B). Compared to the first replica, which remained close to the initial structural model, the dimeric interface in the third replica was smaller but more compact (Fig. 5C and Supplementary Fig. 12). This interface shows closer contacts between opposite charges and a more compact hydrophobic core.

We then calculated distributions of FRET distances for the three replicas (Table 2). For this purpose, we used FRETpredict, an approach that overcomes limitations in AV calculations. FRETpredict systematically takes into account the protein conformational ensemble and accurately models the conformational ensemble of the fluorophore

labels[45]. The predicted values for T-T and T-H/H-T from the replicas R2 and R3 agree quantitatively very well with single-molecule FRET data obtained from fixed-cell and live-cell experiments. This suggests that the InlB-mediated interface remains relatively flexible and enables InlB to fluctuate between a planar conformation to a reversed V shape. The results of integrating atomistic MD simulations and smFRET show that in situ, the MET:InlB dimer deviates from the crystal form II organization (Fig. 5D).

## Discussion

Despite the key importance of plasma membrane-receptor-mediated cellular events, the structural dynamics of receptor activation in situ are still poorly understood. Determining the receptor oligomeric state by solution methods of soluble fragments or structural methods like X-ray crystallography or single-particle cryo-electron microscopy (cryo-EM) may lead to incomplete answers. For example, the binding of the GAS6 ligand to the soluble ECD of the receptor tyrosine kinase (RTK) AXL results in a 1:1 complex, although a 2:2 AXL:GAS6 complex is most likely formed on the cell surface[46]. As a further example, crystallography revealed conflicting structural models for signaling active complexes of fibroblast growth factor (FGF) bound to the ECD of its receptor FGFR, another RTK[47]. Therefore, in vitro studies often need to be complemented by analysis of receptors in the membrane or a membrane-like environment, like lipid nanodiscs or amphipols that are increasingly used in single-particle cryo-EM. Even then, receptor complexes may require chemical cross-linking to avoid dissociation during purification[48].

In this study, we present a comprehensive mechanistic analysis of the early activation steps of the MET receptor upon binding of the bacterial ligand InlB. Two MET:InlB dimer structures (form I, PDB 2UZX; and form II, PDB 2UZY) with contrasting orientations of InlB were proposed[24].

As is often the case, deriving the actual quaternary structure from these crystal structures is difficult, because contacts in the crystal may represent either mere crystal packing contacts or physiologically relevant protein-protein interactions. Physiological dimers are usually $C_2$ symmetric. As both form I and form II of the (MET:InlB₃₂₁)₂ complex

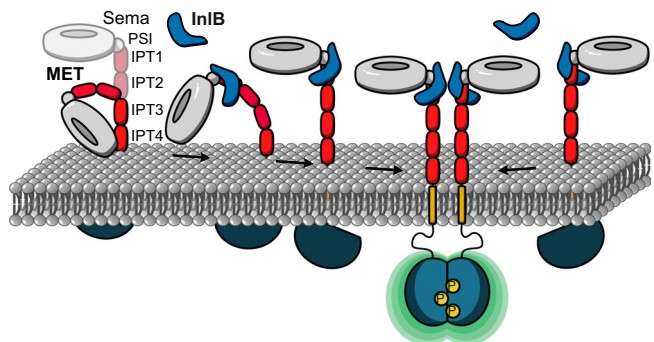

**Fig. 6 | Mechanistic model of MET receptor activation upon InlB binding.** In the ligand-free state, the ectodomain of MET shows pronounced flexibility, while the binding of InlB stabilizes an extended conformation. The extended conformation facilitates the association of two MET:InlB complexes to form the signaling-active (MET:InlB)$_2$ complex.

have $C_2$ point group symmetry, this criterion did not help in deciding between both assemblies[49]. Another criterion used to distinguish crystal-packing contacts from evolved protein-protein interactions is the size of the interface. Form I of the MET:InlB dimer has a substantially larger interface than form II (3700 Å$^2$ and 1400 Å$^2$, respectively). This is presumably the main reason why the PISA server suggests form I to assemble a stable 2:2 complex in solution, whereas it predicts form II to exist only as a 1:1 complex. Experimentally, we never observed dimerization of the MET ectodomain by InlB$_{321}$ in solution[50]. Therefore, we initially suggested that InlB clusters MET into larger complexes in the plasma membrane without the formation of discrete 2:2 complexes[24]. Later, we hypothesized that form II could represent a biologically relevant 2:2 complex, although it neither is predicted nor observed to be a stable 2:2 complex in solution[51].

The currently available crystal and cryo-EM structures resolve MET only down to IPT2 due to the considerable flexibility of the MET stalk region. Hence, the structural organization of the presumed (MET:InlB)$_2$ complex in the plasma membrane of cells, including the entire MET stalk region, remained unclear.

To access the structural organization of membrane receptors in situ, we established an integrative structural biology workflow by complementing structural insights with single-molecule experiments, modeling, and MD simulations. Based on these findings, we propose a mechanistic model for the early activation of the MET receptor by the bacterial ligand InlB (Fig. 6).

Extensive equilibrium MD simulations show that the ectodomain of MET is in a conformational equilibrium between a compact and an extended structure. In the compact conformation, the ectodomain bends significantly, bringing the Sema domain into direct contact with the membrane headgroup region (Fig. 1F and Supplementary Fig. 2C). The ectodomain of integrins, the α-subunit of which is structurally similar to MET, adopts in their inactive form a bent conformation on the membrane surface[52] (PDB 3K71), which closely resembles the compact conformation explored by the MET ectodomain in our simulations (see R2 in Fig. 1F).

The binding of InlB favors the extended conformation of the MET ectodomain (see Fig. 1G), which we hypothesize is the signaling-competent monomer. The extended conformation enables back-to-back interactions between two internalin that facilitate the formation of MET dimers. When bound to MET, InlB bridges the Sema domain and the stalk of MET, forcing the structure into a stiff conformation characterized by an angle of about $\theta_b = 135°$. Notably, this is the same angle formed in MET:HGF monomers[17], even though the HGF-mediated dimer organization differs significantly from the internalin mediated one. The 135°-conformation appears to be mechanistically critical for receptor activation.

Informed by crystal structures[51,53], we designed a single-molecule FRET experiment to determine the in situ structure of the (MET:InlB)$_2$ complex (later referred to as in situ dimer). Our in situ smFRET measurements now unequivocally show that on cells, discrete 2:2 MET:InlB complexes do form and they also inform about possible structures of these 2:2 complexes in the native environment of the cell membrane. The smFRET data clearly ruled out the form I assembly in cells under physiological conditions. At the same time, distance information retrieved from smFRET experiments was not in quantitative agreement with those inferred from the crystal structure of form II.

Reconciling this discrepancy required sampling the structural dynamics of the complex with MD simulations and using an accurate model of the smFRET experiment[45]. Three independent MD replicas showed local rearrangements of the dimer (Fig. 5B). The third replica (R3) of the form II crystal structure, which aligns well with the T-T and T-H/H-T distances from smFRET experiments, explains how the native (MET:InlB)$_2$ dimer attains stability: the dimeric interface was smaller but more compact, with closer contacts between opposite charges and a more compact hydrophobic core (Fig. 5C and Supplementary Fig. 12). The back-to-back arrangement of the InlBs in the identified dimer structure is in accordance with cell-based receptor activation assays using InlB variants intended to block assembly of either form I or form II[49] and the increased activity achieved when cross-linking InlB proteins in a similar configuration[51]. The recurrence of form II in a second crystal form[54] further supports our in situ (MET:InlB)$_2$ dimer model. Lastly, the analysis of the MD trajectories of InlB-bound to the MET ectodomain corroborates the reported lower affinity of the IR-Sema interface compared to the LRR-IPT1 interface[24]. In particular, we observed that in 1 out of 3 replicas, this interface dissociates, providing flexibility to the MET stalk. The combination of smFRET experiments and MD simulations elucidated the assembly of the native dimer in situ.

A critical step of our analysis was the accurate determination of distances from smFRET data. Two benchmark studies conducted by the smFRET community demonstrated an achievable precision no greater than 0.2 nm and an accuracy below 0.5 nm in DNA and protein samples[55,56]. The distances obtained from our smFRET analysis were confirmed in MD simulations and allowed the refinement of the structural model of the (MET:InlB)$_2$ dimer. In addition, smFRET analysis reports on the structural flexibility of a protein assembly. Photon distribution analysis (PDA)[43,44] (Supplementary Fig. 11) indicated that the (MET:InlB)$_2$ dimer predominantly adopts a single conformation. The T-H/H-T FRET dataset exhibits a slightly larger differential between experimental and simulated data, stemming from the two potential binding positions for these FRET combinations, while the simulated datasets contain only a single-state population (Supplementary Note 2).

It has been reported that fixation of cells with formaldehyde (FA) might impair the accuracy of FRET results[57–59], while other studies have not observed such interference[42,60]. These observations are typically made if fluorescent proteins are used as reporters since fixation restricts their orientation and flexibility. In order to assess the potential effect of FA fixation in this study, we performed FRET experiments both in fixed and live cells. We observed very similar FRET efficiency values in both cases, which we attribute to a fixed conformation within the (MET:InlB)$_2$ complex and the use of organic fluorophores and short linkers. We did observe some broadening of the FRET efficiency distributions obtained from live-cell experiments that we attribute to shorter integration times in single-particle tracking experiments[32,33,61,62], stronger variations in background, and extended noise levels. In addition, a structural flexibility within the (MET:InlB)$_2$ complex might occur. In conclusion, we present a robust in situ smFRET study of an intact membrane protein complex in both fixed and living cells and could exclude potential artifacts originating from chemical fixation.

A recent single-particle cryo-EM study that required forced MET dimerization by the addition of a leucine-zipper motif reported two

distinct structures for MET dimers bound to the ligands HGF and NK1[17]. Interestingly, one HGF ligand was found to be sufficient to dimerize two MET receptors, by binding to two distinct binding sites of MET. This asymmetric 2:1 MET:HGF complex can bind another HGF and assemble into a 2:2 (MET:HGF)$_2$ complex. In contrast, binding of the NK1 isoform to MET leads to the formation of a symmetric 2:2 (MET:NK1)$_2$ complex, in which the NK1 proteins directly interact in a head-to-tail fashion and form themselves a dimer and that is structurally similar to the (MET:InlB)$_2$ dimer[51,53]. This suggests that (MET:HGF)$_2$ and (MET:InlB)$_2$ dimers are structurally organized in significantly different ways. It will be interesting to understand whether different structural arrangements at the level of the ectodomain propagate to the intracellular domain, potentially mediating alternative downstream events.

While our structural model illustrates key events of MET activation, it also sheds light on exciting questions. The MET ectodomain is glycosylated, but the role played by glycans in its structural dynamics is not understood. In the inactive monomer, the Sema domain is in direct contact with the membrane. Specific interactions between the ectodomain and lipid headgroups could further modulate the conformational equilibrium between the inactive and active monomers.

In summary, our model provides insights into the structural dynamics of monomeric MET and the dynamic interplay between MET and InlB, and provides a useful methodological framework to study receptor activation and dimerization on the plasma membrane. Our study illustrates once more that crystal structures provide excellent working hypotheses but do not always exactly correspond to the conformation of biomolecular complexes in the cell. Integration of in situ single-molecule experiments with dynamical molecular simulations represents a powerful approach to determining the organization of complexes in the cell.

## Methods
### Atomistic molecular dynamics of MET upper ectodomain
We modeled the atomistic upper ectodomain (UniProt P08581-1 sequence numbering, residues 43–657) of the MET receptor in isolation and in complex with the InlB$_{321}$ fragment of the InlB protein starting from the crystallographic structure PDB 2UZY[24]. We modeled the missing residues (UniProt P08581-1 sequence numbering, 92–110, 151–155, 206–209, 302–311, 378–383, 398–406, 411– 413, 628–633) with the MODELLER[63] plug-in implemented on UCSF Chimera[64]. We added 8 A2 N-glycans (di-sialylated, bi-antennary complex-type N-glycans) in both models on experimentally determined N-glycosylation sites (UniProt P08581-1 sequence numbering, residue 45, 106, 149, 202, 399, 405, 607, 635)[65].

We solvated the systems using CHARMM-GUI[66–68] in combination with GROMACS[69] as MD engine. We minimized the systems using the steepest-descent algorithm for 5000 steps and performed a 125-ps-long NVT (particle amount, volume, and temperature are kept constant) equilibration using the Nose-Hoover thermostat with a reference temperature of 310 K ($\tau_t = 1$ ps). We simulated the systems in the NPT (particle amount, pressure and temperature are kept constant) ensemble for 3 μs each using the Charmm36m forcefield[70,71] with TIP3 water model, a reference temperature of 310 K ($\tau_t = 1$ ps, V-rescale thermostat), a reference pressure of 1 bar ($\tau_p = 5$ ps, Parrinallo-Rahman barostat) and a NaCl concentration of 0.15 M. For both the Van der Waals (Verlet) and the Coulomb forces (Particle Mesh Ewald), we used a $r_{cut-off} = 1.2$ nm. We used a 2 fs timestep. We used GROMACS 2021.3[69] for the MET:InlB system and GROMACS 2021.4[69] for the isolated MET model.

### Definition of θ angle
We computed θ as defined by two vectors describing the relative orientation of the Sema and the IPT1 domains. The first vector

connects the centers of mass of two groups of atoms on the upper and lower sides of the Sema domain (182–200 and 464–479); the second vector connects two groups of atoms at the opposite sides of the IPT1 cylinder (561–657 and 655–657). We calculated the value of θ from the simulated atomistic trajectories using custom-written code in the Python packages NumPy[72] and MDAnalysis[73].

### Atomistic molecular dynamics of the entire MET ectodomain
We initially produced two models of the entire ectodomain of the MET receptor. The first one spans residues 43-930 (UniProt P08581-1 sequence numbering). The second one spans residues 1-930 and contains an N-terminal loop of 42 amino acids and a disulfide bond between CYS26 and CYS584 on IPT1, which could affect the ectodomain's structural dynamics. For the first model we started from the equilibrated upper ectodomain model as described above. In the absence of an experimental structure, we built on the AlphaFold prediction[21] of the IPT2, IPT3, and IPT4 domains. We used the MET receptor structure reported on the AlphaFold database[21] corresponding to the UniProt entry P08581. We trimmed the IPT2-IPT3-IPT4 fragment (UniProt P08581-1 sequence numbering, residues 658-930) of the predicted structure and connected it to the upper ectodomain models (MET and MET:InlB) using UCSF Chimera[64]. For the second model, we started from the cryo-EM structure PDB 7MO7[17], which consists of the SEMA, PSI, IPT1, and IPT2 domains. In the absence of an experimental structure, we modeled the IPT3-IPT4 by connecting the IPT3-IPT4 fragment from the AlphaFold prediction[21] to the cryo-EM model using UCSF Chimera[64].

In the crystal structure of the InlB-bound MET, the 43-residue N-terminal loop is not resolved, unlike in the cryo-EM structure, which includes it but does not show InlB binding interfaces. To rule out modeling biases in the MET:InlB$_{321}$ complex, we used the experimentally determined crystal structure, expecting that the presence of the disulfide bond would not significantly affect the system due to the observed stability of the upper ectodomain when simulated bound to InlB$_{321}$. To account for the impact of N-glycosylation, we included 11 A2 glycans in all models (UniProt P08581-1 sequence numbering, residue 45, 106, 149, 202, 399, 405, 607, 635, 785, 879, 930)[65].

In addition, the comparison of the isolated ectodomain structural dynamics in the two models showed that the N-terminal loop and the CYS26-CYS584 bond are consistent with a very compact conformation and do not significantly affect the extended-compact dynamics of the ectodomain (Supplementary Fig. 13). To enable an optimal comparison between the isolated and InlB-bound models, we focused our simulations on the crystal-based models, which are also slightly smaller and enable longer simulated trajectories and therefore better statistics. To quantify the extension of the ectodomain, we exploited the radius of gyration ($R_g$). We calculated the $R_g$ using the MDAnalysis function *radius_of_gyration*[73].

We prepared the systems using CHARMM-GUI solution builder[66–68] in combination with GROMACS[69] as the MD engine. We minimized the systems using the steepest-descent algorithm for 5000 steps (for the MET in isolation with an N-terminal loop, we performed two minimization runs) and performed a 125-ps-long NVT equilibration using the Nose-Hoover thermostat with a reference temperature of 310 K ($\tau_t = 1$ ps). We simulated the systems in the NPT ensemble for 2.5 μs each using Charmm36m forcefield with GROMACS 2021.4 with TIP3 water model, a reference temperature of 310 K (V-rescale thermostat, $\tau_t = 1$ ps), a reference pressure of 1 bar (Parrinallo-Rahman barostat, $\tau_p = 5$ ps) and a NaCl concentration of 0.15 M. For both the Van der Waals (Verlet) and the Coulomb forces (Particle Mesh Ewald), we used a $r_{cut-off} = 1.2$ nm. We chose a 2 fs timestep. To quantify the convergence of the systems, we compared the average RMSD of the replicas for the two systems.

## Atomistic molecular dynamics of the entire MET ectodomain inserted in a membrane

On the basis of the models of the entire MET ectodomain isolated and bound to InlB, we produced the corresponding models inserted in a membrane. The models span residues 43-985 of the MET receptor (UniProt P08581-1 sequence numbering). To build the models, we attached the entire MET ectodomain models to the TMD domain as predicted by AlphaFold 2[21], in the absence of an experimental structure. The AlphaFold 2 prediction of the TMD is in perfect agreement with the secondary structure information reported in UniProt (UniProt P08581-1). We then used CHARMM-GUI[66–68] to produce a standard 1-palmitoyl-2-oleoyl-sn-*glycero*-3-phosphocholine (POPC) bilayer and UCSF Chimera[64] to achieve the final assemblies. We solvated the system at a NaCl concentration of 0.15 M using GROMACS 2021.5[69]. We minimized, equilibrated, and ran the production runs of the models using the Charmm36m forcefield and GROMACS 2022.4[69]. The minimization was achieved using a steepest-descent algorithm. For the equilibration, we ran 2 rounds of NVT equilibration using the Berendsen thermostat ($\tau_t = 1$ ps) at a reference temperature of 310 K. We then ran 4 additional rounds of NPT equilibration using the same settings for the temperature coupling and a semi-isotropic pressure coupling using the Berendsen barostat ($\tau_p = 5$ ps) at a reference pressure of 1 bar. To achieve a proper equilibration of the membrane and protein system, we progressively reduced the force constants of the restraints applied to the lipids and protein backbone. In all runs, we used a $r_{cut-off} = 1.2$ nm for both the Van der Waals (Verlet) and the Coulomb forces (Particle Mesh Ewald). We used a timestep of 1 fs for equilibration runs and of 2 fs for production.

## Quasi-atomistic molecular dynamics of the entire MET ectodomain

We produced quasi-atomistic coarse-grained (CG) models of MET in isolation and in complex with InlB$_{321}$ using the Charmm-GUI Martini maker web server[74] selecting MARTINI 3 forcefield[26] with elastic network. We removed the interdomain elastic bonds and, to maintain the ligand in place, we applied harmonic restraints between the InlB and the binding interfaces on both SEMA and IPT1 domains using a distance threshold of 2 nm between the backbone beads (Supplementary Fig. 3A). We performed both operations using custom python scripts. Following the same procedure detailed above, we obtained a coarse-grained model of the MET in isolation. For this model, we removed the interdomain elastic bonds using a custom Python notebook and modified the bond type from standard type 1 to 6.

We then used the *insane.py* script[75] to solvate the systems. We used a NaCl concentration of 0.15 M with neutralizing ions. Using GROMACS[69], we ran two rounds of minimization using the steepest descent algorithm. The first ran until convergence, the second for 6000 steps. Then, we ran the NPT equilibration at a reference temperature of 310 K (V-rescale thermostat, $\tau_t = 1$ ps) and a reference pressure of 1 bar (Berendsen barostat, isotropic coupling type, $\tau_p = 5$ ps). We simulated the systems in the NPT ensemble at a reference temperature of 310 K (V-rescale, $\tau_t = 1$ ps) and a reference pressure of 1 bar (Parrinello-Rahman barostat, isotropic coupling type, $\tau_p = 12$ ps). Coulombic interactions were treated with PME and a cut-off of 1.1 nm, as for van der Waals interactions. We used a time step of 20 fs. For the MET:InlB$_{321}$ complex system, we used GROMACS 2022.6[69], for the MET in isolation, we used GROMACS 2020.5[69].

## Molecular dynamics of the (MET:InlB)$_2$ upper ectodomain dimer

We created an atomistic model of the MET:InlB dimer in the form II as reported by PDB 2UZY[76] by aligning two copies of the NVT equilibrated model described in the "Atomistic molecular dynamics of MET upper ectodomain" section. We then simulated 3 atomistic replicas of this model. Firstly, we solvated the system with TIP3P water using GROMACS[69]. We minimized the systems using the steepest-descent

algorithm for 5000 steps and performed a 125-ps-long NVT equilibration using the Nose-Hoover thermostat with a reference temperature of 310 K ($\tau_t = 1$ ps). We simulated each replica in the NPT ensemble for 0.6 μs using the Charmm36m forcefield, a reference temperature of 310 K ($\tau_t = 1$ ps, V-rescale thermostat), a reference pressure of 1 bar ($\tau_p = 5$ ps, Parrinello-Rahman barostat) and a NaCl concentration of 0.15 M. For both the van der Waals (Verlet) and the Coulomb forces (Particle Mesh Ewald), we used a $r_{cut-off} = 1.2$ nm. We chose a 2 fs timestep. We used GROMACS 2021.4.

## Prediction of FRET distances from atomistic MD simulations

We predicted smFRET distances from the atomistic MD simulations of the MET:InlB form II dimer using the Python package FRETpredict[45]. This method uses rotamer libraries of FRET dyes superimposed to protein structures or trajectories to predict the FRET efficiency distributions[45]. It considers the structural dynamics of the FRET dyes and their linkers. We adapted the tutorial jupyter notebooks (downloaded at https://github.com/KULL-Centre/FRETpredict) for our experiments. As the rotamer libraries for our dye pair were not available, we performed ~1.2 μs atomistic MD simulations for each dye in the solution. To correctly reproduce the dynamics of the dyes, we used the CHARMM-DYES forcefield, which includes optimized parameters for our FRET dye pair[77]. We employed CHARMM-DYES combined with the same solvation conditions used in the simulations of the dimer model. The CHARMM-DYES forcefield did not include parameters for the maleimide ring used in experiments nor the thioester bond between the linker and the cysteine residue. Therefore, to approximate the experimental linker length and flexibility, we used a C4 linker where the first two dihedrals were disregarded to account for the stiffness of the missing ring while retaining almost the same bond length. We used the calculated rotamer libraries to perform the FRET efficiency prediction. To account for the position of the linker as attached to the S atom of the cysteine, we set the offset for the rotamer placement on the Cγ atom of the corresponding residue. The FRET signal produced by the dye pair in T-H/H-T arrangements is indistinguishable in the experiments due to the isotropic character of the dimerization process after treatment. We, therefore, averaged the predictions of distributions of T-H and H-T. The position of the residues on the InlB enabled us to use the $k^2$ approximation, which allowed us to obtain the efficiency values predicted using the static regime calculation[45]. We calculated the distributions of the T-T and T-H/H-T efficiencies for the 3 different replicas. We assessed the local convergence of the replicas by calculating the RMSD of the Cα and Cβ atoms in the InlB-InlB dimer. All FRET predictions were obtained on a locally equilibrated fragment of each replica. We estimated the standard deviation of the FRET predictions by applying bootstrapping on the time series of the predicted FRET signal. To perform this task, we used the *pandas* function *Series* in combination with the *sample* function.

## Passivation and functionalization of 8-well chambers and coverslips for single-molecule experiments

For *d*STORM measurements and smFRET experiments in fixed cells, 8-well chambers (SARSTEDT AG & Co. KG, Nümbrecht, Germany) were prepared by plasma cleaning with nitrogen for 10 min at 80% power and 0.3 mbar using a Zepto B plasmacleaner (Diener Electronic GmbH, Ebhausen, Germany). Next, 2 μL of 0.8 mg/mL RGD-grafted poly-L-lysine-graft-(polyethylene glycol) (PLL-PEG-RGD) (prepared according to Harwardt et al. [32]) were diluted in 23 μL of ddH$_2$O per well. The chambers were incubated with the PLL-PEG-RGD solution at 37 °C for 1 h before drying in a sterile bench at room temperature for 2 h. Cells were seeded on the same day that the PLL-PEG-RGD coating was prepared.

For live-cell smFRET experiments, round coverslips (25 mm diameter, 0.17 mm thickness, VWR International GmbH, Radnor, PA, USA)

were sonicated (S30H Elmasonic, Elma electronic GmbH, Pforzheim, Germany) in isopropanol for 20 min at 35 °C, then washed three times with ddH$_2$O, and dried with nitrogen. The coverslips were plasma cleaned with nitrogen for 10 min at 80% power and 0.3 mbar. Then, 10 µL of 0.8 mg/mL PLL-PEG-RGD solution was distributed between two coverslips and incubated for 1.5 h in 10 cm cell culture dishes (Greiner, Bio-One International GmbH, Kremsmünster, Austria) at room temperature. When the incubation was finished, the coverslips were rinsed with ddH$_2$O and separated carefully. After drying with nitrogen, they were placed in 6-well plates (Greiner, Bio-One International GmbH) and stored under argon, sealed with parafilm, at −20 °C for up to 2 weeks.

## Cell culture

The human osteosarcoma cell line U-2 OS (# 300364, CLS Cell Lines Service GmbH, Eppelheim, Germany) was cultivated in high glucose DMEM/nutrient mixture F-12 (DMEM/F12) (# 11320033, Gibco, Life Technologies, Thermo Fisher Scientific, Waltham, MA, USA) with 1% GlutaMAX (# 35050-038, Gibco), penicillin (1 unit/mL), streptomycin (1 µg/mL; Gibco, Life Technologies) and 10% fetal bovine serum (FBS) (# 35-079-CV, Corning Inc., Corning, NY, USA) at 37 °C and 5% CO$_2$ in an automatic CO$_2$ incubator (Model C 150, Binder GmbH, Tuttlingen, Germany). The cervix carcinoma cell line HeLa (# ACC 57, DSMZ, Braunschweig, Germany), the hepatocellular carcinoma cell line Huh 7.5 (DKFZ Heidelberg), and the astrocytoma cell line U-251 (# 300385, CLS Cell Lines Service GmbH) were cultivated in high glucose DMEM (# 11574486, Gibco) with 1% GlutaMAX (Gibco) and 10% FBS (Corning Inc.) and the gastric adenocarcinoma cell line 23132/87 (# ACC 201, DSMZ, Braunschweig, Germany) was cultivated in RPMI medium (# 11875093, Gibco) with 1% GlutaMAX (Gibco) and 10% FBS (Corning Inc.) as described above. Cells were split every 3-4 days.

For *d*STORM experiments, 23132/87, HeLa, Huh 7.5, U-2 OS, and U-251 cells were seeded onto PLL-PEG-RGD-coated 8-well chambers in the respective medium with penicillin (1 unit/mL) and streptomycin (1 µg/mL; Gibco, Life Technologies) at densities between $0.5 \times 10^4$ to $2.5 \times 10^4$ cells/well. For smFRET measurements in fixed cells, U-2 OS cells were seeded onto PLL-PEG-RGD-coated 8-well chambers (300 µL cell suspension with $1 \times 10^4$ cells/well) and grown with penicillin (1 unit/mL) and streptomycin (1 µg/mL; Gibco, Life Technologies) for 3 days. For smFRET measurements in living cells, U-2 OS cells were seeded onto PLL-PEG-RGD-coated coverslips (diameter 25 mm, VWR International GmbH) in 6-well culture plates with 2 mL cell suspension of $3 \times 10^4$ cells/well and grown in growth medium with penicillin (1 unit/mL) and streptomycin (1 µg/mL; Gibco, Life Technologies) for 3 days. For western blots, $2 \times 10^6$ cells were seeded in 10 cm cell culture dishes and incubated at 37 °C and 5% CO$_2$ for 3 days.

## *d*STORM experiments

**Immunofluorescence of MET.** Two days after seeding, the medium of the cells was exchanged against serum-free medium and the cells were grown for one further day. For immunofluorescence, cells were washed once with 1x phosphate-buffered saline (PBS, # 14190-094, Gibco) pre-warmed to 37 °C. Cells were fixed with prewarmed 4% methanol-free formaldehyde (# 28908, Thermo Scientific) in 1x PBS for 10 min. After washing thrice with 1x PBS, samples were blocked with a blocking buffer (BB) containing 5% (w/v) bovine serum albumin (BSA) (# A7906-50G, Sigma-Aldrich, St. Louis, MO, USA) in 1x PBS for 1 h at room temperature with gentle shaking. The primary antibody (goat anti-MET, # AF276, lot CMQ0720032, R&D Systems, USA) was diluted in BB to a final concentration of 2 µg/mL and incubated for 2 h at room temperature with gentle shaking. After the incubation, the cells were washed three times with 1x PBS. The Alexa Fluor 647 rabbit@goat secondary antibody (2 µg/mL in BB, # A-21446, Invitrogen, Thermo Scientific, Germany) was added to the cells and incubated for 1 h at room temperature with gentle shaking. For negative controls, cells

were incubated with secondary antibody only, without primary antibody. After washing three times with 1x PBS, the cells were fixed for 10 min with 4% methanol-free formaldehyde in 1x PBS. Gold beads with a diameter of 100 nm (# A11-100-NPC-DIH-1-25, lot M1139, Nanopartz, USA) were used as fiducial markers. The gold beads stock solution was vortexed shortly and then sonicated for 10 min. A 1:5 dilution was prepared with 1x PBS and sonicated again for 10 min. The dilution of the fiducial markers was added to the cells and incubated for 15 min. Finally, cells were washed three times with 1x PBS and stored in 0.05% (w/v) NaN$_3$ (# S2002-25G, Sigma-Aldrich) in 1x PBS at 4 °C until further use.

**dSTORM imaging.** *d*STORM imaging was performed in an imaging buffer containing β-mercaptoethylamine (MEA, # 30078-25 G, Sigma-Aldrich) as a reducing agent and glucose oxidase/catalase as an oxygen scavenging system. The imaging buffer containing 10% (w/v) glucose (# G7528-1KG, Sigma-Aldrich), 100 mM MEA, 50 U/mL glucose oxidase (# G2133-50KU, Sigma-Aldrich), and 5000 U/mL catalase (# C3155-50MG, Sigma-Aldrich) in 1x PBS was prepared freshly before the measurements. The pH was adjusted to 8 with 1 M NaOH (# S8045-1KG, Sigma-Aldrich).

*d*STORM measurements were performed with an N-STORM microscope (Nikon, Japan) equipped with an oil-immersion objective (100x Apo TIRF, NA 1.49, Nikon) and an EMCCD camera (DU-897U-CS0-#BV, Andor Technology, UK). µManager (v1.4.22)[78] and NIS elements (version 4.30.02, Nikon, Germany) were used for acquisition control. An image size of 256 px x 256 px to 320 px x 320 px was chosen depending on the size of the cell. A 647 nm laser was used for the excitation of Alexa Fluor 647, and a 405 nm laser for reactivation. The laser intensity of the 647 nm was set to 0.4 kW/cm². The 405 nm laser was adjusted as necessary to obtain a regular blinking (0−22 mW/cm²). The camera settings were as follows: exposure time 50 ms, EM gain 200, preamp gain 3, frame transfer on, and film lengths 30,000 frames. All measurements were performed with total internal reflection fluorescence (TIRF) illumination. For each cell line, at least three independent experiments were performed.

**Data analysis.** *d*STORM movies were analyzed with the Picasso software[79]. The point-spread functions of single molecules were localized with Picasso Localize using the following parameters: box side length 7 px, min net gradient 60,000, EM gain 200, baseline 216, sensitivity 4.78, quantum efficiency 0.95, pixel size 157 nm, and maximum-likelihood estimation. Drift correction was performed in Picasso Render either with RCC or by picks using the gold beads as fiducial markers. Next, localizations were filtered in Picasso Filter for their standard deviations in the x and y direction (0.6–1.6 px). The experimental localization precision was determined in Picasso using the nearest neighbor analysis (NeNA)[80]. Localizations of the same binding event were linked using six times the NeNA value (or a maximum value of 0.45 px) and 5 dark frames. The number of receptor clusters was determined using the density-based spatial clustering and application with noise (DBSCAN) algorithm[81]. A radius of two times the NeNA value (or a maximum value of 0.15 px) and a minimum number of 10 localizations were set. The cluster number divided by the cell area (determined in Fiji) yielded the MET receptor cluster density.

## Single-molecule FRET with alternating laser excitation

**Generation of site-specifically labeled InlB variants.** InlB$_{321}$ (comprising amino acids 36-321 of the full-length InlB) was produced by fusing it with a cleavable glutathione-S-transferase (GST) protein using the tobacco etch virus (TEV) protease[37]. To prevent the formation of unwanted disulfide bonds, a C242A mutation was introduced. This mutation does not affect the binding of MET[19]. Two InlB variants were generated, and the respective mutation K64C (H) or K280C (T), as well as the C242A mutation, were introduced into the pETM30 vector using

the QuikChange® mutagenesis kit (Stratagene)[24]. The plasmids of the InlB variants are available upon request. *Escherichia coli* BL21-Codon-Plus(DE3)-RIL cells transformed with the vector were cultured in lysogeny broth (LB) medium supplemented with kanamycin and chloramphenicol at 37 °C until reaching an optical density at 600 nm of 0.6. Following induction with 0.1 mM isopropyl βD-1-thiogalacto-pyranoside, $InlB_{321}$ variants were expressed overnight with shaking at 20 °C. The cells were harvested through centrifugation and lysed. After centrifugation, the lysate was applied to a glutathione sepharose affinity matrix equilibrated in 1x PBS. The resin was washed with 1x PBS and TEV protease cleavage buffer (50 mM Tris-HCl pH 8.0, 0.5 mM EDTA, 20 mM NaCl, 1 mM DTT) and then resuspended in TEV cleavage buffer. TEV protease and dithiothreitol (DTT) were added and incubated at room temperature overnight for cleaving $InlB_{321}$ from the GST tag. $InlB_{321}$ was purified further using anion exchange chromatography. Specifically, $InlB_{321}$ was loaded onto a Source Q 15 column equilibrated with 20 mM Tris buffer pH 7.5 and eluted with a linear gradient of salt concentration (up to 300 mM NaCl). For labeling $InlB_{321}$ with fluorophores, freshly purified $InlB_{321}$ was used immediately after elution and stored under nitrogen. Tris(2-carboxyethyl)phosphine (TCEP) was added to reach a final concentration of 0.5 mM. 100 µg of ATTO 647N maleimide (# AD 647N, ATTO-TEC, Siegen, Germany) or CyTM3b Mono maleimide (#PA13101, GE Healthcare, Frankfurt, Germany), respectively, were dissolved in 5 µL of dry dimethylformamide (DMF). A 3-fold molar excess of TCEP was added to the $InlB_{321}$ protein, followed by the addition of a 3-fold molar excess of dye-maleimide conjugate relative to the protein. The mixture was incubated for 1 h at room temperature in the dark. To remove the unbound fluorophore, a PD10 desalting column (GE Healthcare) equilibrated with 1x PBS was used. Protein purity was evaluated by Coomassie-stained SDS-PAGE. Protein concentrations and degrees of labeling (DOL) were determined spectrophotometrically by measuring the absorbance at 280 nm and 644 nm using a NanoPhotometer (Implen GmbH, Munich, Germany). Labeled $InlB_{321}$ was stored in the dark at − 20 °C.

**Sample preparation.** Three days after seeding, U-2 OS cells were rinsed with 400 µL prewarmed, serum-free DMEM/F12 and then starved for 2 h in serum-free DMEM/F12 at 37 °C and 5% $CO_2$. For ligand stimulation, Cy3B- and ATTO 647N-labeled $InlB_{321}$ variants ($InlB_{321}$-H or $InlB_{321}$-T) were added to a final concentration of 5 nM per InlB variant. As controls, only one $InlB_{321}$ variant was used. Cells were incubated with the ligand for 15 min at 37 °C. To examine background fluorescence signals, the cells were first incubated 5 min with 200 nM single-chain Fv (scFv) fragment of a previously published antibody (107_A07), then 5 nM of T-$InlB_{321}$ variants were added, and incubated for 15 min. The scFv 107 is a derivative of the anti-MET antibody 107_A07[41] and was obtained from Ardis S.r.l (Pavia, Italy). The protein was produced in *Pichia pastoris* and purified by affinity and size exclusion chromatography to yield a fully monomeric protein. Immediately after stimulation, cells were washed once using 200 µL/well of prewarmed 0.4 M sucrose solution in 1x PBS (diluted from 10x stock, # 14200067, Gibco), followed by fixation for 15 min at room temperature using a solution consisting of 4% formaldehyde (Thermo Scientific) and 0.01% glutaraldehyde (# G5882, Sigma-Aldrich) in 0.4 M sucrose and 1x PBS. Subsequently, cells were rinsed three times using 300 µL 1x PBS.

To reduce photobleaching during single-molecule measurements, an oxygen scavenging buffer (300 µL/well) was employed which was prepared freshly before each measurement: glucose oxidase from *Aspergillus niger* type VII (0.009 U/µL; Sigma-Aldrich), catalase from bovine liver (594 U/mL; Sigma-Aldrich), glucose (0.083 M; Sigma-Aldrich), and Trolox (1 mM; Sigma-Aldrich)[82,83].

**Setup and data acquisition.** Single-molecule FRET measurements were performed on a home-built total internal reflection fluorescence (TIRF) microscope based on an Olympus IX-71 inverted microscope (Olympus Deutschland GmbH, Hamburg, Germany). The excitation light was provided by two lasers (637 nm, 140 mW OBIS and 561 nm, 200 mW Sapphire, both Coherent Inc., Santa Clara, CA, USA). Both laser beams were collinearly superimposed using a dichroic mirror (H 568 LPXR super flat, AHF Analysentechnik AG, Tübingen, Germany). An acousto-optical tunable filter (AOTF; AOTFnC-400.650-TN, AA Opto-Electronic, Orsay, France) selected the excitation light, which alternated between 561 nm and 637 nm. The required timing was achieved by means of two digital counter/timer and analog output devices (NI PCI-6602 and NI PCI-6713, National Instruments, Austin, TX, USA). To spatially overlay both lasers and clean the beam profiles, the lasers were coupled by a fiber collimator (PAF-X-7-A, Thorlabs, Dachau, Germany) into a single-mode optical fiber (P5-460AR-2, Thorlabs) and subsequently re-collimated to a diameter of 2 mm (60FC-0-RGBV11-47, Schäfter & Kirchhoff, Hamburg, Germany). The collinear beams were then directed to a 2-axis galvo scanner mirror system (GVS012/M, Thorlabs) where electronic steering, controlled by an in-house Python script, allowed switching between wide-field illumination, steady-state and circular TIRF, and HILO (highly inclined and laminated optical sheet) modes of operation. The excitation beams were then directed through two telescope lenses (AC255-050-A-ML and AC508-100-A-ML, Thorlabs), which focused the beams onto the back focal plane of the objective (UPlanXApo, 100x, NA 1.45, Olympus Deutschland GmbH). In a filter cube, which directs the beam into the objective, two clean-up and rejection bandpass filters together with a dichroic mirror were installed (Dual Line Clean-up ZET561/640x, Dual Line rejection band ZET 561/640, Dual Line beam splitter zt561/640rpc, AHF Analysentechnik AG). A nosepiece stage (IX2-NPS, Olympus Deutschland GmbH) provided z-plane adjustment and minimized drift during the measurements.

Fluorescence emission was collected through the same objective and passed the dichroic mirror toward the detection path. An Optosplit II (Cairn Research Ltd, UK) was used to split the fluorescence light around 643 nm into two channels using a beam splitter together with two bandpass filters (H643 LPXR, 590/20 BrightLine HC, 679/41 BrightLine HC, AHF Analysentechnik AG). The two spatially separated donor and acceptor channels were simultaneously detected on an EMCCD camera (iXon Ultra X-10971, Andor Technology Ltd, Belfast, UK). The setup achieved a total magnification of 100x, resulting in a pixel size of 159 nm. The µManager software[78] captured 1000 frames with the following settings: exposure time 100 ms, EM gain 150, pre-amp gain 3x, readout rate 17 MHz, image size 512 × 256 pixels, and activated frame transfer. Bright-field images of the cells were taken after each measurement. The excitation laser wavelengths were alternated between 561 nm and 637 nm for a duration of 100 ms each. The laser intensities in the sample were 10.7 W/cm² and 73.0 W/cm², respectively. For each sample, four independent experiments were performed. To align both channels, daily measurements of 100 nm TetraSpeck™ microspheres (# T7279, Invitrogen, Thermo Fisher Scientific, Waltham, MA, USA) on coverslips were conducted for 100 frames without alternating lasers.

**Data analysis.** Single-molecule FRET movies were analyzed using the iSMS software[39]. The 561 nm and 637 nm excitation channels were aligned with the default settings of the autoalign ROIs tool. FRET pairs were detected, averaging the intensity of all 1000 frames. Initially, we considered every donor and acceptor position as a potential FRET pair. We manually selected FRET traces based on two criteria: an increase in donor intensity upon photobleaching of the acceptor and single-step photobleaching in both the donor and acceptor channels to ensure that only a single donor-acceptor fluorophore pair was present.

Selected smFRET intensity traces were corrected in iSMS for donor emission leakage into the acceptor channel ($\alpha$), acceptor direct excitation by the donor excitation laser ($\delta$), and different detection

efficiencies and quantum yields of donor and acceptor $(\gamma)$[39]. The iSMS software determined $\alpha$, $\delta$, and $\gamma$ trace-wise. The mean correction factors were applied to the data within iSMS. In addition, we manually calculated the $\beta$-correction factor which normalizes for different excitation intensities and cross-sections of donor and acceptor.

$$\beta = \frac{avg(I_{AA})}{avg(\gamma I_{DD})} \quad (1)$$

where $I_{AA}$ represents the emission intensity of directly excited acceptor and $I_{DD}$ denotes the donor emission intensity from direct excitation. The FRET efficiencies and stoichiometries were determined following a published protocol[56] and computed with OriginPro (OriginLab Corporation, Northampton, MA, USA):

$$E = \frac{I_{DA} - \alpha I_{DD} - \delta I_{AA}}{\gamma I_{DD} + (I_{DA} - \alpha I_{DD} - \delta I_{AA})} \quad (2)$$

$$S = \frac{\gamma I_{DD} + (I_{DA} - \alpha I_{DD} - \delta I_{AA})}{\gamma I_{DD} + (I_{DA} - \alpha I_{DD} - \delta I_{AA}) + \frac{1}{\beta} I_{AA}} \quad (3)$$

Here, $I_{DA}$ is the acceptor intensity when the donor is excited. The calculated FRET efficiencies were histogrammed and the distribution for each condition was fitted with a Gaussian distribution to obtain the FRET efficiency for the respective condition. The distances $R$ between donor and acceptor fluorophores were calculated from these FRET efficiencies.

$$R = R_0 \cdot \sqrt[6]{\frac{1}{E} - 1} \quad (4)$$

Here, $R_0$ is the fluorophore-pair-specific Förster radius.

$$R_0 = 0.211 \cdot \sqrt[6]{\kappa^2 \cdot n^{-4} \cdot \phi_D \cdot J(\lambda)} \quad (5)$$

For the orientation factor $\kappa^2$, free rotation of the fluorophores was assumed, therefore $\kappa^2 = 2/3$. The refractive index $n$ of the imaging solution was measured to be 1.34. The quantum yield $\phi_D$ of the donor is given by the fluorescence decay rate $k_F$ and the fluorescence lifetime $\tau_L$.

$$\phi_D = k_F \cdot \tau_L \quad (6)$$

Cy3B $\phi_D$ was calculated from the fluorescence decay rate (0.239 ns$^{-1}$; calculated using quantum efficiency and lifetime from Cooper et al. [84]) and the lifetime of the donor determined by time-correlated single photon counting (TCSPC) for each Cy3B-labeled InlB$_{321}$ variant (T variant: $\tau_L = 2.6$ ns, H variant: $\tau_L = 2.5$ ns). The fluorescence lifetimes were measured using a PicoHarp 300 system (Picoquant, Berlin, Germany) in combination with a pulsed 485 nm laser for excitation. Finally, $J(\lambda)$ represents the overlap integral of Cy3B emission and ATTO 647N absorption ($5.8 \cdot 10^{15}$ M$^{-1}$ cm$^{-1}$ nm$^4$; FPbase[85]). The Förster radius $R_0$ calculated for the T variant is 6.34 nm and for the H variant 6.30 nm. The errors of the FRET efficiencies and donor-acceptor distances were calculated according to Agam et al. and Hellenkamp et al. [55,56].

The photon distribution analysis (PDA) was conducted using software provided by the group of Claus Seidel (https://www.mpc.hhu.de/software/pda). The underlying model for this analysis is derived from Antonik et al. [43,44]. Emissions from both, donor and acceptor, under donor excitation, were converted into a 2D histogram. Subsequently, this histogram was imported into the Tatiana software for further analysis. The fitting of the ratio between donor and acceptor emission followed Antonik et al.'s approach, employing free fit parameters, except for two fixed parameters: the number of limited width states and dynamic states, both set at 1.

## Live-cell single-molecule FRET

**Sample preparation.** The coverslips with cells were mounted into custom-built coverslip holders. 500 μL pre-warmed live cell imaging solution (# A59688DJ, Gibco) was added to the cells. Then the cells were cooled to room temperature for 10 min. Directly after that, an oxygen scavenging buffer (0.009 U/μL glucose oxidase from *Aspergillus niger* type VII, 594 U/mL catalase from bovine liver, 0.083 M glucose), 1 mM Trolox[62], and the fluorophore-labeled ligands were added to the cells and incubated for 5 min. Ligand concentrations were 30 nM for both, Cy3B and ATTO 647N-labeled InlB$_{321}$ variants (T-T, H-T, or H-H). Each well of cells was measured for a maximum of 30 min at room temperature to guarantee cell viability.

**Data acquisition.** Live-cell smFRET measurements were performed according to the previously published smFRET recovery after photobleaching (smFRET-RAP) method[42]. The SPT movies were acquired at the same setup as the smFRET measurements on fixed cells. The cells were first photobleached in TIRF illumination for 5 to 10 min with the 561 nm (67.7 W/cm$^2$) and 637 nm (552.1 W/cm$^2$) lasers until the background of the basal membrane disappeared, followed by a 1 min recovery period without illumination. In order to verify the photobleaching process, the cells were captured for 500 frames with 561 nm (19.8 W/cm$^2$) and 637 nm (76.5 W/cm$^2$) lasers before and after photobleaching. Then, 4000 frames were captured only with 561 nm excitation (19.8 W/cm$^2$) for FRET observation. The following settings were used for data acquisition with the μManager software[78]: exposure time 40 ms, EM gain 200, preamp gain 3x, readout rate 17 MHz, image size 512 × 256 pixels, and activated frame transfer. A bright field image was recorded after each measurement. For each sample, more than three independent experiments were performed. To align both channels, 100 nm TetraSpeck™ microspheres were recorded for 100 frames with 561 nm and 637 nm lasers at each measurement day.

**Data analysis.** For data analysis, the donor and acceptor channels were aligned using the TetraSpeck™ measurements with the default settings of iSMS using the autoalign ROIs tool. Donor and acceptor channels were separated based on the determined ROIs. The separated frame series of the two channels were analyzed with u-track[86]. Single-molecule signals were localized using the Gaussian Mixture-model fitting using the default settings except for the PSF radius, which was derived from the data in 10 iterations. 2D tracking was performed with a gap closure time of 3 frames and a minimum trajectory length of 20 frames. smFRET analysis was performed using the Matlab-based smCellFRET software[42]. The potential smFRET traces were extracted and manually selected based on the following criteria: an increase in donor intensity upon photobleaching of the acceptor or a decrease in donor intensity upon an increase of the acceptor intensity. The selected FRET traces were summarized in SPARTAN[87] and exported for further data processing. The FRET distribution histograms of T-T and H-T combinations were generated in OriginPro. The FRET efficiency was calculated as introduced for the smFRET measurements in fixed cells. For the H-H combination, no FRET traces were found.

**Donor-acceptor distance estimation by AV simulation.** To estimate the distances between donor and acceptor in the (MET:InlB$_{321}$)$_2$ complex for different InlB variants, we applied accessible volume (AV) simulations[88]. AV simulations predict the allowed average distances between donor and acceptor dyes. It was achieved by the FRET Positioning and Screening (FPS) software[28] using the parameters summarized in Table 3. The FRET-averaged distances are shown in Fig. 2.

**Table 3 | AV simulation parameters used for donor-acceptor distance estimation within the (MET:InlB$_{321}$)$_2$ complex**

|  | Linker length [Å] | Linker width [Å] | R$_1$ [Å] | R$_2$ [Å] | R$_3$ [Å] |
|---|---|---|---|---|---|
| ATTO 647N maleimide | 21.0 | 4.5 | 7.15 | 4.5 | 1.5 |
| Cy3B maleimide | 18.5 | 4.5 | 3.4 | 8.2 | 3.0 |

The linker is simplified as a cylinder model; the length and width represent the height and radius of the cylinder. The dye is simulated as an ellipsoid using the 3AV model. The radii R$_1$, R$_2$, and R$_3$ describe the dye ellipsoid. Linker and dye dimensions were taken from Klose et al.[92] for Cy3B maleimide and from Claus Seidel (Heinrich Heine University Düsseldorf) for ATTO 647N maleimide.

**Western blots**. U-2 OS cells were rinsed with 5 mL serum-free DMEM-F12 per dish and then starved in 10 mL serum-free DMEM-F12 medium for at least 8 h at 37 °C. The starvation minimizes the signal activation in cells due to the activating ligands for MET that may be contained in the serum, and synchronizes the cells, leading them into a similar phase of the cell cycle. After starvation, the cells were stimulated at 37 °C for 15 min with 2 mL of 5 nM InlB$_{321}$ variant (Supplementary Table 1) or 1 nM HGF (# 100-39H, PeproTech, Hamburg, Germany) in serum-free DMEM-F12. For the resting condition, cells were only treated with serum-free DMEM-F12 medium for 15 min. Then cells were rinsed with 10 mL ice-cold 1x PBS and kept for 2 min on ice. PBS was then removed and 80 μL of lysis buffer (Triton X-100 (# T8787-50ML, Sigma-Aldrich) 1%, Tris-HCl (pH 7.4, # 15568-025, Invitrogen) 50 mM, NaF (# 71519-100 G, Sigma-Aldrich) 1 mM, NaCl (# 71376-1KG, Sigma-Aldrich) 150 mM, Na$_3$VO$_4$ (# 450243, Sigma-Aldrich) 1 mM, EDTA (# AM9260G, Invitrogen) 1 mM, and ¼ cOmplete Mini, EDTA-free protease inhibitor tablet, # 11836153001, Roche for 10 mL) were added per dish and incubated on ice for at least 30 s. The cells were scraped thoroughly to one corner of the dish and collected in ice-cold 1.5 mL tubes. When all the samples were collected on ice, they were shaken at 750 rpm using a thermoshaker and 4 °C for 5 min and centrifuged at 12,400 × $g$ and 4 °C for 20 min, the supernatants were collected in new tubes and stocked shortly on ice. The concentration of total proteins was determined with the BCA Protein Assay Kit (# K813-2500, VWR International GmbH). According to the total protein amount in each sample, 1 M DTT (# 43819-1 G, Sigma-Aldrich), 5x loading dye (Tris-HCl (pH 6.8) 250 mM, sodium dodecyl sulfate (SDS, # L6026-50G, Sigma-Aldrich) 8% (w/v), bromophenol blue (# 114391, Sigma-Aldrich) 0.1% (w/v), glycerol (# G5516-100ML, Sigma-Aldrich) 40% (v/v)), and ddH$_2$O were mixed so that the protein amount was 30 μg protein and the final concentrations were 100 mM DTT and 1x loading dye. The samples were stored at −20 °C until further use. For sodium dodecyl sulfate-polyacrylamide gel (SDS-PAGE), samples were heated to 95 °C for 5 minutes before cooling down on ice. Each pocket of the SDS-PAGE gel (# 4561094, Bio-Rad, Hercules, CA, United States) was filled with a 35 μL sample or 6 μL PageRuler (# 26617, Thermo Fisher Scientific). Gel electrophoresis was performed in running buffer (Tris base (# 154563-1KG, Sigma-Aldrich) 25 mM, glycine (# G8898-500G, Sigma-Aldrich) 192 mM, SDS 3.46 mM in ddH$_2$O) at 170 V for around 45 min. The protein was transferred from the gel to the western blot with an iBlot gel transfer system (# IB1001, Invitrogen, Thermo Fisher Scientific) for 7 min. Each blot was blocked with 10 mL 5% (w/v) nonfat dry milk (# 9999S, Cell Signaling Technology, Danvers, MA, USA) in TBST buffer (25 mM Tris base, 150 mM NaCl, and 0.05% (v/v) Tween-20, pH 7.6) for 1 h at room temperature. After blocking, the blots were washed 3 times with TBST buffer and shaken gently with 5 mL primary antibody (rabbit anti-MET, #4560, lot 2, Cell Signaling Technology, 1:1000 dilution, or rabbit anti-pMET, #3077, lot 9, Cell Signaling Technology, 1:1000 dilution, and rabbit anti-actin, #ab14130, lot 487755, abcam, Cambridge, UK, 1:10000 dilution) in 5% (w/v) bovine serum albumin in TBST at 4 °C overnight. The excess of primary antibodies was removed by washing 3 times with TBST. The blots were incubated with 10 mL secondary antibody (goat anti-rabbit HRP, #111-035-003, Jackson ImmunoResearch, West Grove, PA, USA, 1:20000 dilution in 5% (w/v) BSA in TBST) at room temperature for 3 h. Afterward, the blots were rinsed 4 times with TBST and one time with TBS (25 mM Tris base and 150 mM NaCl, pH 7.6). For every wash step, the blot was incubated for at least 5 min. The blots were visualized by a CHEMI-only chemiluminescence imaging system (VWR International GmbH). The quantitative analysis was done using the open-source Fiji software (NIH, USA)[89].

### Reporting summary
Further information on research design is available in the Nature Portfolio Reporting Summary linked to this article.

## Data availability
Single-molecule imaging data have been deposited in the EMBL BioImaging Archive under accession code S-BIAD1347[90]. The MD simulation data and parameter files are freely available on Zenodo (https://doi.org/10.5281/zenodo.14007780)[91]. Source data are provided in this paper.

## Code availability
The analysis code for the MD simulation data is freely available on Zenodo (https://doi.org/10.5281/zenodo.14007780).

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

## Acknowledgements

We thank Björn Hellenkamp for helpful suggestions for the AV simulations, Sören Doose and Markus Sauer for access to their TCSPC spectrometer, Suren Felekyan for providing the PDA software, Johanna Rahm for coding support for the PDA analysis, Alexandra Kaminer for supporting the analysis of live-cell smFRET data, and Wenjun Wang for Matlab coding support for live-cell smFRET analysis. M.H., Y.L., and

M.S.D. acknowledge funding by the Deutsche Forschungsgemeinschaft (CRC1507: Membrane-associated Protein Assemblies, Machineries, and Supercomplexes, project ID 450648163; INST 161/778-1 FUGG). H.H.N. acknowledges funding by the Deutsche Forschungsgemeinschaft (grant NI 694/3-1). S.M.A., G.J.H., and R.C. acknowledge the support of Goethe University Frankfurt, the Frankfurt Institute of Advanced Studies, the CRC 1507: Membrane-Associated Protein Assemblies, Machineries and Supercomplexes (Deutsche Forschungsgemeinschaft), the LOEWE Center for Multiscale Modeling in Life Sciences of the state of Hesse, and the International Max Planck Research School on Cellular Biophysics. R.C. acknowledges the support of Bayreuth University. S.M.A., G.J.H., and R.C. acknowledge computational resources and support by the Center for Scientific Computing of the Goethe University, and the Gauss Center for Supercomputing e.V. (www.gauss-centre.eu) for funding this project by providing computing time on the GCS Supercomputer JUWELS at Jülich Supercomputing Center (JSC).

## Author contributions

M.H. and R.C. designed the project. H.H.N. designed the FRET study and, together with D.H. and D.M.F., provided labeled proteins. P.F. assisted with cell culture. H.D.B. designed the optical setup for smFRET. Y.L. and M.S.D. performed microscopy experiments and, together with M.H., analyzed and interpreted the data. S.M.A. and G.J.H. performed MD simulations and, together with R.C., analyzed and interpreted the data. L.I. and H.d.J. synthesized the single-chain Fc fragment. M.H., R.C., Y.L., S.M.A., H.H.N., and M.S.D. discussed the data and wrote the manuscript.

## Funding

## Competing interests

L.I. and H.d.J are listed as co-founders of the company Ardis S.r.l. All other authors declare no competing interests.
