## [Transparent Peer Review file · Nature Communications]

Single-molecule imaging and molecular dynamics simulations reveal early activation of the MET receptor in cells

Corresponding Author: Professor Mike Heilemann

Version 0:

Reviewer comments:

Reviewer #1

(Remarks to the Author)

The manuscript by Li et al. investigates the membrane assembly of MET receptors in the presence of the pathogenic protein internalin B. The authors have used a combination of in-cell smFRET, atomistic, and quasi-atomistic MD simulations to predict MET activation. They also propose a mechanistic model for MET receptor activation. Cryo-EM structures are available for the MET receptors, however, what is still needed is an understanding of the dynamics of the protein as well as structural insight in a cellular environment. This study is able to fill in this important gap and provide insight into the binding mode of InIB to MET in cells. The smFRET experiments done in situ is novel. However, the data is limited and is not sufficient to provide the experimental evidence to back up the simulations and the proposed mechanism. Several controls are also needed to validate the smFRET data. Additionally, data is from fixed cells and this is a significant concern.

Major concerns:

1. The proposed mechanism is largely based on MD simulations, the limited smFRET is insufficient to validate the proposed mechanism. Additional experimental data such as mutations, cross-linking or other biochemical evidence is needed to validate the model proposed.
2. Controls are needed to validate the smFRET data. Donor only data to show fraction of spots showing only two step photobleaching is needed. Mutations at MET that block InIB binding to MET to show no background fluorescence is needed. Mutations in MET that block dimerization to show monomeric InIB binding with single photobleaching steps.
3. The validation of form 1 versus form 2 is based on AV predictions (Table 1). The variability in prediction versus experiment is 0.62 to 0.82 (which is not surprising as a number of factors can shift FRET to higher or lower efficiencies). But given the same reasoning it is also likely that the FRET of 0.258 may be shifted to lower efficiencies such that it is not observed. The lower FRET efficiencies are hard to observe as the number of photons in acceptor channel may be very low and similar to what one would expect for donor alone. This would be mean that the complex may have both form 1 and form 2 as possibilities. This needs to be addressed based on prior or additional data that support the presence of a single form of the complex.
4. In the sample preparation step, the authors have used 4% paraformaldehyde to fix the cells. PFA acts as a fixer by crosslinking proteins. How does this affect the dimerization and dynamics? Could this type of cross-linking favor one form over the other?

Minor revisions:

5. The figure legend in Fig. 1C says "Side and front view of the closed and open conformations". What do the authors mean by this? Is the monomeric MET referred to as 'Closed'? Not sure if the usage of the terms 'open/close' is appropriate here. Fig. 1C looks confusing.
6. Page 3, line 4 "Signaling of MET is initiated by binding of the physiological ligand HGF16,17, its natural isoform NK1 17,18 or the bacterial ligand InIB" needs a reference for InIB signaling.
7. Consider adding subscript on the angles θ_1 & θ_2 to make it clear on which angle is being described for the rest of the reading
8. Figure 2B & C – consider changing the colors of the white spheres to represent the yellow and green color of the head and tail for easier understanding as the numbers and background currently cover most of the important parts of the structure
9. Figure 2B & C – include the PDBs in the description for B and C instead of at the end as it is unclear which form comes

from which PDB based on the figure only

Reviewer #2

(Remarks to the Author)

The manuscript "Single-molecule imaging and molecular dynamics simulations reveal early activation of the MET receptor in situ" by Li et al. explores the activation mechanisms of a single-pass transmembrane protein, the human tyrosine kinase growth factor receptor MET, using a combination of structural methods, in silico modeling and molecular dynamics simulations, and single-molecule Förster resonance energy transfer (smFRET) techniques.

Despite the technical sophistication of the experimental approaches applied to the study of the MET receptor, the work fails to provide a seamless connection between the structural biology/FRET studies with an actual description of MET activation in situ, because none of the studies is conducted in a membrane environment, except for the FRET section in a cellular system, which, however, is undertaken on fixed cells.

The statement in the Discussion section: "The combination of smFRET experiments and MD simulations elucidated the assembly of the native dimer in situ" is therefore not supported by the data presented. In the opinion of this reviewer, elimination of this and similar statements would improve the quality of the paper. Additional evidence would be needed to support the original claim.

Results

To investigate the early stages of the activation process, the authors initially applied atomistic and quasi-atomistic molecular dynamics simulations to mimic the interaction of *Listeria monocytogenes* (a pathogen that targets MET to initiate host cell invasion) with the ectodomain of MET obtained from the crystallographic data (PDB entry: 2UZY), comparing the MD of the isolated ectodomain with the ectodomain in the presence of the *Listeria* invasion protein, InIB. The results of this part of the study showed that InIB binding controls the overall conformation of the MET ectodomain, rigidifying it, and thereby stabilizing the receptor in a conformation that promotes dimer formation.

It is somewhat confusing to read at the end of the fifth paragraph of the Results section (authors did not paginate the manuscript nor indicated line numbers) that "The receptor maintained an upright conformation perpendicular to the plane of the membrane." As far as this reviewer can see, the "plasma membrane" is depicted as a cylinder in Figures 1F and G and is not included in the atomistic molecular modelling exercise. Thus, this is an inference indirectly supported by proxy to literature information, but not by experimental data in the manuscript.

Next, the authors employed single-molecule FRET (smFRET) to explore the orientation of two InIB molecules with the dimeric form of the MET receptor. They first generate two mutants of InIB carrying a single Cys residue of InIB either at position (K64C) or position 280 (K280C) and label these sites with the fluorescent dyes ATTO647N and Cy3B. smFRET was used to measure intramolecular distances in two MET:InIB dimer structures, the so-called complex forms I and II, respectively. smFRET showed similarities and differences with the structure of the dimer disclosed by crystallographic studies.

FRET studies on a set of cells selected on the basis of the appropriate MET receptor cluster densities integrate the use of total internal reflection microscopy with alternating laser excitation and sophisticated analytical techniques to extract FRET efficiencies and detailed measurements of intramolecular distances, to reach the conclusion that the dimer exhibits limited structural flexibility. The authors deliver very solid information from the smFRET that can be compared with the crystal data, establishing similarities and differences with the values predicted from the crystal models.

In a second round of MD simulations, the authors validate the experimental FRET data of the form II dimer model.

Finally, the authors suggest the use of their methodological scheme to study the activation of other membrane receptors in situ.

Discussion

The authors start by emphasizing a key motivation of their work, already stressed in the Introduction: the importance of studying cell-surface phenomena in the membrane or membrane-like environments as opposed to reductionist in vitro studies. Yet the part of the work dedicated to FRET studies in situ (FRET on fixed cells) constitutes a relatively small section of the total.

The authors state: "To access the structural organization of membrane receptors in situ, we established an integrative structural biology workflow by complementing structural insights with single-molecule experiments, modeling and MD simulations". This is an overstatement, because the smFRET and the MD simulations were not conducted in a membrane-mimicking system.

Material and Methods

The Material and Methods section does not follow the sequence of the Results section, making it quite difficult for the reader to follow a logical thread of the procedures employed.

Otherwise, the methodological section on FRET evidences the expertise of Heilemann's group in this topic, being written in a clear form, with great detail, and including appropriate controls.

Western blots section: For the general reader, please add the rationale behind the starvation of the U-2 OS cells.

Reviewer #3

(Remarks to the Author)

The paper "Single-molecule imaging and molecular dynamics simulations reveal early activation of the MET receptor in situ" by Yunqing Li et al. uses an integrative structural biology approach to investigate the activation mechanism of the human growth factor receptor MET. The study combines computational structural modelling, molecular dynamics (MD) simulations, and single-molecule Förster resonance energy transfer (smFRET) experiments to elucidate the early events in MET activation. MET is a receptor tyrosine kinase that plays a crucial role in cell proliferation, migration, and survival and it's often dysregulated in cancer. It is also targeted by the pathogen *Listeria monocytogenes* through its invasion protein InIB. The authors demonstrate that InIB binding stabilises MET in a conformation that promotes dimer formation. Their smFRET experiments provide insights into the organisation of the MET complex in situ, leading to a refined model of the activation mechanism.

The paper presents a significant advancement in the study of membrane receptor activation, combining state-of-the-art techniques to provide detailed mechanistic insights. The integrative approach not only addresses the limitations of individual methods but also opens new avenues for investigating the structure and dynamics of other receptors in their native cellular environments, which is typically not possible with X-ray crystallography or single-particle cryo-electron microscopy. Given the robustness of the methodology, the relevance of the findings, and the potential for broad application, I recommend the publication of this paper.

Minor point: how accurate and predictive are quasi-atomistic molecular dynamics simulations with Martini 3? Considering the numerous constraints applied to the coarse-grained system (elastic network, harmonic restraints), it is unclear whether the resulting insights are truly predictive and add value to the atomistic simulations. Perhaps it would be more prudent to utilise fully atomistic models also for the largest model, given that the authors demonstrate their alignment with the FRET distance distribution for this system.

Version 1:

Reviewer comments:

Reviewer #1

(Remarks to the Author)

The authors have addressed my concerns.

Reviewer #2

(Remarks to the Author)

The authors have addressed this reviewer's comments and queries. In this reviewer's opinion, it is apt for publication.

Reviewer #3

(Remarks to the Author)

In the revised manuscript, the authors have fully addressed my minor concerns. I also believe that they have satisfactorily addressed the points raised by the other reviewers.

Response to Reviewers

Reviewer #1

The manuscript by Li et al. investigates the membrane assembly of MET receptors in the presence of the pathogenic protein internalin B. The authors have used a combination of in-cell smFRET, atomistic, and quasi-atomistic MD simulations to predict MET activation. They also propose a mechanistic model for MET receptor activation. Cryo-EM structures are available for the MET receptors, however, what is still needed is an understanding of the dynamics of the protein as well as structural insight in a cellular environment. This study is able to fill in this important gap and provide insight into the binding mode of InIB to MET in cells. The smFRET experiments done in situ is novel. However, the data is limited and is not sufficient to provide the experimental evidence to back up the simulations and the proposed mechanism. Several controls are also needed to validate the smFRET data. Additionally, data is from fixed cells and this is a significant concern.

Response: We thank the Reviewer for investing valuable time in reading our manuscript, for the very useful suggestions, and for acknowledging the importance of our work.

We have addressed all comments raised by the Reviewer. In addition, we conducted new experiments (both *in cellulo* and *in silico*) and included these in the revised manuscript.

Major concerns:

1. The proposed mechanism is largely based on MD simulations, the limited smFRET is insufficient to validate the proposed mechanism. Additional experimental data such as mutations, cross-linking or other biochemical evidence is needed to validate the model proposed.

Response: We thank the Reviewer for raising this point and for the helpful suggestions alongside. We performed additional experiments and analyses, which are outlined below. In addition, we discuss complementary experimental data that was published by other groups to support our findings; relevant information is integrated in the respective responses below. We apologize that we did not present this preexisting knowledge clearly enough in the previous version of our manuscript.

2. Controls are needed to validate the smFRET data. Donor only data to show fraction of spots showing only two step photobleaching is needed.

Response: We thank the Reviewer for this suggestion. In response to this suggestion, we performed single-molecule photobleaching experiments to monitor MET receptor dimerization. For this purpose, we used InIB variants labeled only with Cy3B (the donor fluorophore in smFRET experiments) and performed single-molecule imaging experiments in cells treated with H-InIB₃₂₁ or T-InIB₃₂₁. The single-molecule photobleaching experiments were conducted in 30 cells acquired in 4 independent experiments. We observed two-step photobleaching for both InIB variants, Cy3B-H-InIB₃₂₁ and Cy3B-T-InIB₃₂₁ (**Figure R1**). This data is now included in the supporting information as Supplementary Fig. S8.

Figure R1: Two-step photobleaching observed in donor-only labeled samples. U-2 OS cells were treated with Cy3B-labeled InIB variants. Exemplary intensity traces are shown for (A) Cy3B-H-InIB₃₂₁ and (B) Cy3B-T-InIB₃₂₁. The intensity over time is shown for the donor (green) and the acceptor upon donor excitation (light orange) as well as for the acceptor upon acceptor excitation (dark orange). Fluorescence intensity is normalized to 1.

Mutations at MET that block InIB binding to MET to show no background fluorescence is needed.

Response: We thank the Reviewer for this comment. Unfortunately, the InIB:MET interface has so far only been probed by mutations in InIB, but not in MET. Hence, no mutations in MET are known that prevent the binding of InIB. Identifying such mutations and introducing them into U-2 OS cells is technically very challenging and would be beyond the scope of this manuscript.

We developed an alternative approach that can unequivocally address the question of binding specificity and background fluorescence. We decided to use a single chain Fv (scFv) fragment of a previously published antibody (107_A07) that binds an epitope on MET IPT1 overlapping with the InIB binding site (DiCara et al. 2017, DOI 10.1038/s41598-017-09460-2). Thus, pre-incubating cells with the 107_A07 scFv will block InIB binding to MET. Our experiments showed that the scFv fragment efficiently blocked InIB binding and we did not see background fluorescence (**Figure R2**). We now included this negative control in the manuscript (Supplementary Fig. S7). The 107_A07 scFv was kindly provided by Luisa Iamele and Hugo de Jonge (University of Pavia), whom we include as additional authors.

Figure R2: TIRF images of U-2 OS cells labeled with Cy3B- and ATTO 647N-labeled InIB variants in the absence and presence of the 107_A07 scFv fragment. Cells were either directly labeled with both 5 nM Cy3B- and ATTO 647N-T-InIB₃₂₁ variants or were previously incubated with 200 nM scFv before the addition of InIB. Subsequently, the cells were chemically fixed. The brightfield images as well as the fluorescence images upon donor and acceptor excitation are shown. Prior incubation with the 107_A07 scFv fragment inhibited the binding of InIB. Scale bars 10 μ m.

Mutations in MET that block dimerization to show monomeric InIB binding with single photobleaching steps.

Response: So far, no mutations in MET are known that allow the binding of InIB but block MET dimerization. All currently known dimerization contacts are located in the InIB:InIB interface. There may be additional MET:MET contacts, e.g. between IPT domains of the MET stalk, between the MET transmembrane domains, or between its tyrosine kinase domains. However, right now there are no structures clearly showing MET:MET contacts involved in dimerization. Thus one would need to guess which MET residues could eventually contribute to dimerization and mutate them in U-2 OS cells on the off chance. We believe that this technically very challenging and labor intensive experiment would go far beyond what is feasible as an additional control experiment. We also note that the cryo-EM structures of MET dimerized by HGF or NK1 (Uchikawa et al. 2021, Nat. Commun., doi.org/10.1038/s41467-021-24367-3) do not show any direct MET:MET contacts. Instead, all dimerization contacts resolved in these structures are mediated by HGF or NK1 located between two MET molecules.

In addition, we performed single-molecule imaging experiments with the truncated ligand InIB₂₄₁. According to published data (Niemann et al. 2007 Cell, DOI 10.1016/j.cell.2007.05.037, Fig. 5a+b), InIB₂₄₁ binds MET with the same affinity as InIB₃₂₁, but it is at the very least 10x less active. This suggests that we should see InIB₂₄₁ binding to MET, yet with significantly less formation of MET dimers.

The truncated InIB₂₄₁ does not allow to generate pairs of single-cysteine mutants for smFRET experiments. Hence, we conducted single-molecule photobleaching experiments and compared the fraction of (MET:InIB)₂ dimers formed by InIB₂₄₁ and by InIB₃₂₁ (the consistency of single-molecule photobleaching and smFRET in reporting MET dimerization has been shown above (response to comment #2)). The analysis of the imaging data reported a very small fraction of dimers in cells treated with InIB₂₄₁ (for a large number of

cells, zero dimers were found) as compared to cells treated with InIB₃₂₁ (**Figure R3**). Hence, InIB₂₄₁ leads to the formation of a (MET:InIB) monomer.

Figure R3: 2-step photobleaching detected for InIB₂₄₁ and InIB₃₂₁. U-2 OS cells were either incubated with InIB₂₄₁ or InIB₃₂₁ labeled with Cy3B at the T position and subsequently chemically fixed. The intensity traces of visually separate signals were analyzed concerning 1- and 2-step photobleaching and quantified. The 25th percentile (dotted line), the median (dashed line), the 75th percentile (dotted line), and the mean (circle) are displayed. Data points of individual cells are shown (diamonds).

3. The validation of form 1 versus form 2 is based on AV predictions (Table 1). The variability in prediction versus experiment is 0.62 to 0.82 (which is not surprising as a number of factors can shift FRET to higher or lower efficiencies). But given the same reasoning, it is also likely that the FRET of 0.258 may be shifted to lower efficiencies such that it is not observed. The lower FRET efficiencies are hard to observe as the number of photons in acceptor channel may be very low and similar to what one would expect for donor alone. This would mean that the complex may have both form 1 and form 2 as possibilities. This needs to be addressed based on prior or additional data that support the presence of a single form of the complex.

Response: We thank the Reviewer for raising this important point.

Although the predicted and experimental FRET efficiencies differ, the experimental values could only be reconciled with form II: (1) the AV predicted FRET efficiencies for T-T are 0.258 (form I) and 0.620 (form II), and the experimental FRET efficiency was measured as 0.86; (2) the H-H distance would not lead to a detectable FRET efficiency for form II (Table 1), which is in line with our experimental data; (3) previous biochemical studies also support that form 2 is preferred (Ferraris et al. 2010, DOI 10.1016/j.jmb.2009.10.074; Niemann et al. 2012, DOI 10.1002/pro.2142).

The remaining differences in the FRET efficiency between form II (0.620) and experiment (0.86) are beyond an acceptable error in a smFRET experiment given all the corrections we conducted. Hence, this was the starting point for us to perform MD simulations on the InIB:MET dimer. The results of these simulations showed that *in situ*, a structure that resembles form II occurs, yet with structural differences (see Figure 5 in the manuscript). Calculating the predicted FRET efficiency from this new MD-derived dimer model overlaps very well with the experimental FRET data (Table 2). This is further supported by the new live cell FRET data.

With the above discussion in mind, we would like to answer the remaining concerns by the Reviewer: (1) can both form I and II co-exist, e.g. through a not-detected FRET efficiency of 0.258 (form I)? We exclude this possibility for several reasons: (i) First, Figure 4A clearly

shows one population only, centered at 0.86. As a single-molecule method, we would expect to see data points at $E < 0.5$, which were not detected. (ii) Our method is sensitive to very low FRET efficiencies. We analyzed the H-H smFRET data again for overlapping signal in the donor and acceptor channel, and by removing the constraint of anti-correlated signal (to reflect for very low I_{DA} signal). Since we used alternating laser excitation (ALEX), we can distinguish a very low FRET efficiency from a donor-only population by using the stoichiometry information. For the H-H pair, we do see a FRET population with an efficiency of 0.028 (matching form II) that is distinguished from the donor-only peak. We show this new analysis in **Figure R4A** for the T-T pair and in **Figure R4B** for the H-H pair.

Figure R4: smFRET experiments of (MET:InIB)₂ in U-2 OS cells using alternating excitation. smFRET E, S-histogram for (A) Cy3B-T-InIB₃₂₁, InIB T-Cy3B and ATTO 647N-T-InIB₃₂₁ including donor-only population (N = 56 smFRET traces from 28 cells), and (B) Cy3B-H-InIB₃₂₁ and ATTO 647N-H-InIB₃₂₁ (N = 41 smFRET traces from 22 cells).

4. In the sample preparation step, the authors have used 4% paraformaldehyde to fix the cells. PFA acts as a fixer by crosslinking proteins. How does this affect the dimerization and dynamics? Could this type of cross-linking favor one form over the other?

Response: We thank the Reviewer for this comment and for the chance to clarify our experimental procedure.

We incubated living cells with the InIB variants and chemically fixed the cells afterward. Thus, the fixation step happens after dimerization, and the impact on the structure of the complex should be minimal.

However, to further exclude artifacts introduced by fixation, we performed single-molecule FRET experiments in live cells. For this purpose, we incubated living cells with InIB₃₂₁ labeled at different positions with Cy3B and ATTO 647N (T/T, H/T, H/H) for an expected high-FRET, mid-FRET, and no-FRET signal. The live cell smFRET experiments were performed employing an established method (smFRET-RAP) and software (Asher et al. 2021, DOI 10.1038/s41592-021-01081-y).

Live cell measurements yielded very similar FRET efficiencies for the T/T and H/T pairs as obtained previously in fixed cells (**Figure R4**). This data is included in the revised manuscript as Figure 4DE and Supplementary Fig. 9.

Figure R4: Live-cell single-molecule FRET of (MET:InIB₃₂₁)₂ dimers in U-2 OS cells. Living cells were labeled with both 5 nM Cy3B- and ATTO 647N-T-InIB₃₂₁ variants and single FRET pairs were analyzed and their FRET efficiency determined. 564 (T-T) and 757 (H-T) smFRET traces from over 20 cells in 3 independent measurements were analyzed.

Minor revisions:

5. The figure legend in Fig. 1C says “Side and front view of the closed and open conformations”. What do the authors mean by this? Is the monomeric MET referred to as ‘Closed’? Not sure if the usage of the terms ‘open/close’ is appropriate here. Fig. 1C looks confusing.

Response: We thank the Reviewer for spotting this and changed the caption accordingly.

6. Page 3, line 4 “Signaling of MET is initiated by binding of the physiological ligand HGF16,17, its natural isoform NK1 17,18 or the bacterial ligand InIB” needs a reference for InIB signaling.

Response: We thank the Reviewer for pointing out these missing references. We now added references for InIB signaling (Shen et al. 2000, DOI 10.1016/s0092-8674(00)00141-0; Banerjee et al. 2004, DOI 10.1111/j.1365-2958.2003.03968.x).

7. Consider adding subscript on the angles θ_1 & θ_2 to make it clear on which angle is being described for the rest of the reading

Response: To improve the wording we included the subscript “b” to indicate the θ angle when MET is bound to InIB.

8. Figure 2B & C – consider changing the colors of the white spheres to represent the yellow and green color of the head and tail for easier understanding as the numbers and background currently cover most of the important parts of the structure.

Response: We thank the Reviewer for this helpful comment. The white spheres were originally thought to represent the fluorophores (either donor or acceptor) attached to the respective cysteine residues. We agree with the Reviewer that the numbers covered most of the structure. Therefore, we removed the numbers from the images and instead added a legend indicating the different distances. We also removed the white spheres for better visibility of the structure (**Figure R5**, new Figure 2 in the manuscript).

Figure R5: InIB₃₂₁ site-specifically labeled variants in two possible MET:InIB dimer structures differing by the orientation of the MET:InIB monomers. (A) Crystal structure of InIB₃₂₁ with the two cysteine mutation sites marked in orange and green (PDB 1H6T). (B) Form I (PDB 2UZ_X) and (C) form II (PDB 2UZ_Y) of the (MET:InIB₃₂₁)₂ complex. The distances between the different combinations of InIB₃₂₁ variants are indicated. MET is shown in gray and InIB in blue.

9. Figure 2B & C – include the PDBs in the description for B and C instead of at the end as it is unclear which form comes from which PDB based on the figure only

Response: The manuscript was changed accordingly.

Reviewer #2

The manuscript “Single-molecule imaging and molecular dynamics simulations reveal early activation of the MET receptor *in situ*” by Li et al. explores the activation mechanisms of a single-pass transmembrane protein, the human tyrosine kinase growth factor receptor MET, using a combination of structural methods, *in silico* modeling and molecular dynamics simulations, and single-molecule Förster resonance energy transfer (smFRET) techniques.

Response: We thank the Reviewer for their time and the careful reading of the manuscript.

Despite the technical sophistication of the experimental approaches applied to the study of the MET receptor, the work fails to provide a seamless connection between the structural biology/FRET studies with an actual description of MET activation *in situ*, because none of the studies is conducted in a membrane environment, except for the FRET section in a cellular system, which, however, is undertaken on fixed cells.

The statement in the Discussion section: “The combination of smFRET experiments and MD simulations elucidated the assembly of the native dimer *in situ*” is therefore not supported by the data presented. In the opinion of this reviewer, elimination of this and similar statements would improve the quality of the paper. Additional evidence would be needed to support the original claim.

Response: We thank the Reviewer for challenging us on the *in situ* nature of our study. We beg to differ with the Reviewer’s statement “none of the studies is conducted in a membrane environment, except for the FRET section”, given the fact that the FRET experiments on cells constitute an extensive and central set of experiments. The statement that “the FRET section [...] is undertaken on fixed cells” misses the point that our experiments allow receptor dimerization to proceed in the native membrane environment of living cells before fixation. We previously showed that InIB can dimerize MET in living cells before fixation, but not if cells are fixed before the addition of InIB (Dietz et al. 2013, DOI doi.org/10.1186/2046-1682-6-6).

To further strengthen our results, we performed single-molecule FRET experiments in living cells which yielded very similar results; this data is included in the revised manuscript (Figure 4DE, Figure S9). For a detailed description on these experiments, we refer to our response to a similar query raised by Reviewer #1.

Results

To investigate the early stages of the activation process, the authors initially applied atomistic and quasi-atomistic molecular dynamics simulations to mimic the interaction of *Listeria monocytogenes* (a pathogen that targets MET to initiate host cell invasion) with the ectodomain of MET obtained from the crystallographic data (PDB entry: 2UZY), comparing the MD of the isolated ectodomain with the ectodomain in the presence of the *Listeria* invasion protein, InIB. The results of this part of the study showed that InIB binding controls the overall conformation of the MET ectodomain, rigidifying it, and thereby stabilizing the receptor in a conformation that promotes dimer formation.

It is somewhat confusing to read at the end of the fifth paragraph of the Results section (authors did not paginate the manuscript nor indicated line numbers) that “The receptor maintained an upright conformation perpendicular to the plane of the membrane.” As far as this reviewer can see, the “plasma membrane” is depicted as a cylinder in Figures 1F and G and is not included in the atomistic molecular modelling exercise. Thus, this is an inference indirectly supported by proxy to literature information, but not by experimental data in the manuscript.

Response: We thank the Reviewer for this important remark. We now included new data from atomistic MD simulations of the monomeric MET receptor inserted in a plasma membrane. For further details, please see below the MD-related answers. We also thank the Reviewer for challenging us to be more precise in our language. The new MD simulations in the plasma membrane revealed that the entire MET ectodomain keeps an upright but tilted conformation on the membrane. We removed any reference to the “perpendicular” organization from the new version of the manuscript.

Next, the authors employed single-molecule FRET (smFRET) to explore the orientation of two InIB molecules with the dimeric form of the MET receptor. They first generate two mutants of InIB carrying a single Cys residue of InIB either at position (K64C) or position 280 (K280C) and label these sites with the fluorescent dyes ATTO647N and Cy3B. smFRET was used to measure intramolecular distances in two MET:InIB dimer structures, the so-called complex forms I and II, respectively. smFRET showed similarities and differences with the structure of the dimer disclosed by crystallographic studies.

FRET studies on a set of cells selected on the basis of the appropriate MET receptor cluster densities integrate the use of total internal reflection microscopy with alternating laser excitation and sophisticated analytical techniques to extract FRET efficiencies and detailed measurements of intramolecular distances, to reach the conclusion that the dimer exhibits limited structural flexibility. The authors deliver very solid information from the smFRET that can be compared with the crystal data, establishing similarities and differences with the values predicted from the crystal models.

Response: We thank the Reviewer for the kind assessment of our smFRET experiments.

In a second round of MD simulations, the authors validate the experimental FRET data of the form II dimer model.

Finally, the authors suggest the use of their methodological scheme to study the activation of other membrane receptors *in situ*.

Discussion

The authors start by emphasizing a key motivation of their work, already stressed in the Introduction: the importance of studying cell-surface phenomena in the membrane or membrane-like environments as opposed to reductionist *in vitro* studies. Yet the part of the work dedicated to FRET studies *in situ* (FRET on fixed cells) constitutes a relatively small section of the total.

Response: While it is true that the section dedicated to FRET studies occupies a relatively small portion of the manuscript in terms of text and figures, we would like to emphasize that the technical challenges to overcome, and the amount of manpower and resources invested into obtaining these results were substantial. The complexity and challenges associated with conducting FRET experiments *in situ* required optimization of experimental conditions, extensive data collection, and rigorous analysis. The accuracy obtained relied largely on site-specifically labeled and functional ligands. The smFRET experiments are fundamental to the conclusions we draw and underpin the importance of studying these phenomena in more biologically relevant environments.

smFRET studies conducted in cells are still relatively rare in the field. Due to the complexity of such experiments, they are less represented in the literature compared to *in vitro* studies. For the revision of this manuscript, we conducted new experiments and now report smFRET experiments in living cells, and obtain very similar results as in fixed cells (please see response to reviewer #1).

The authors state: “To access the structural organization of membrane receptors *in situ*, we established an integrative structural biology workflow by complementing structural insights with single-molecule experiments, modeling and MD simulations”. This is an overstatement, because the smFRET and the MD simulations were not conducted in a membrane-mimicking system.

Response: We thank the Reviewer for their valuable input that improved our understanding of the receptor dynamics.

The smFRET experiments were conducted directly in cells. For this purpose, living cells were treated with InIB ligands, followed by fixation and single-molecule imaging. In the frame of this revision, we also conducted smFRET experiments in living cells, which supported the results obtained in fixed cells (new data shown in Figure 4DE and Supplementary Fig. 9) (please see also the responses to similar questions raised by Reviewer #1).

To further support our findings from the MD simulations of the entire ectodomain in solution, we conducted additional simulations to examine the dynamic structural ensembles of monomeric MET within a membrane environment. The large size of the resulting system

made these simulations computationally very demanding, and we could not simulate trajectories long enough to obtain a statistically converged sampling. Nevertheless, the results of these new simulations are consistent with our previous conclusions: InIB favors an extended conformation of the monomeric MET stalk (new Supplementary Fig. 12). Additionally, the simulations of MET on the membrane highlight that the kink between IPT3 and IPT4, observed also in solution (Figure 1G), induced the monomeric MET to assume a tilt angle with respect to the membrane plane (new Supplementary Fig. 12).

Material and Methods

The Material and Methods section does not follow the sequence of the Results section, making it quite difficult for the reader to follow a logical thread of the procedures employed.

Response: We thank the Reviewer for pointing this out. We changed the order of the subsections in the Material and Methods section and now follow the sequence of the Results section.

Otherwise, the methodological section on FRET evidences the expertise of Heilemann's group in this topic, being written in a clear form, with great detail, and including appropriate controls.

Response: We thank the Reviewer for acknowledging our work.

Western blots section: For the general reader, please add the rationale behind the starvation of the U-2 OS cells.

Response: We thank the Reviewer for this suggestion. We added a sentence to the manuscript explaining the reason for starvation.

Reviewer #3

The paper "Single-molecule imaging and molecular dynamics simulations reveal early activation of the MET receptor in situ" by Yunqing Li et al. uses an integrative structural biology approach to investigate the activation mechanism of the human growth factor receptor MET. The study combines computational structural modelling, molecular dynamics (MD) simulations, and single-molecule Förster resonance energy transfer (smFRET) experiments to elucidate the early events in MET activation. MET is a receptor tyrosine kinase that plays a crucial role in cell proliferation, migration, and survival and it's often dysregulated in cancer. It is also targeted by the pathogen *Listeria monocytogenes* through its invasion protein InIB. The authors demonstrate that InIB binding stabilises MET in a conformation that promotes dimer formation. Their smFRET experiments provide insights into the organisation of the MET complex in situ, leading to a refined model of the activation mechanism.

The paper presents a significant advancement in the study of membrane receptor activation, combining state-of-the-art techniques to provide detailed mechanistic insights. The integrative approach not only addresses the limitations of individual methods but also opens new avenues for investigating the structure and dynamics of other receptors in their native cellular environments, which is typically not possible with X-ray crystallography or single-particle cryo-electron microscopy.

Given the robustness of the methodology, the relevance of the findings, and the potential for broad application, I recommend the publication of this paper.

Response: We thank the Reviewer for the kind assessment of our work and for the time spent reading the manuscript.

Minor point: how accurate and predictive are quasi-atomistic molecular dynamics simulations with Martini 3? Considering the numerous constraints applied to the coarse-grained system (elastic network, harmonic restraints), it is unclear whether the resulting insights are truly predictive and add value to the atomistic simulations. Perhaps it would be more prudent to utilise fully atomistic models also for the largest model, given that the authors demonstrate their alignment with the FRET distance distribution for this system.

Response: We thank the Reviewer for raising this point and allowing us to explain our reasoning better. The motivation to use the quasi-atomistic Martini 3 force field to investigate the structural dynamics of the receptor further was two-fold. On the one hand, the lower computational cost enabled us to collect more sampling for more replicas, letting us collect more statistics. On the other hand, recapitulating the observation that the binding of Internalin-B blocks the upper ectodomain in a specific conformation and that this then affects the rest of the ectodomain showed us that the mechanism is robust and does not rely on subtle interactions.

The Reviewer is also correct in asking how informative the quasi-atomistic simulations are, given that we use some information extracted from the atomistic ones to prepare them. Martini 3 requires, in fact, the application of an elastic network to maintain the protein's tertiary structure. Crucially, we applied elastic networks only **within** each of the domains composing the MET ectodomain, and never between them. The domains in MET's ectodomain are very rigid, as one would expect, and our atomistic simulations also show this. We learn from the Martini 3 simulations that what mostly matters to reproduce the overall structural dynamics is considering the few amino acids constituting the very short loops connecting all the various domains. In this way, we obtained a structural model where all domains were free to reorganize with respect to each other, conditioned on the flexibility of the loops. In summary, the only information we used from the atomistic simulations to parametrize the Martini simulations are the specific binding mode of Internalin and that each domain is very rigid.

Our data also revealed a striking concordance in the overall structural dynamics of MET between the atomistic and quasi-atomistic models. This high level of agreement, despite the fundamentally different descriptions of the interactions, underscores the suitability of our 'mechanical' model. This model, which posits that Internalin primarily controls MET's structure through steric interactions that propagate along the ectodomain, proved effective in understanding the impact of ligand binding for this system.

In response to the Reviewers' suggestions, we also conducted atomistic MD simulations of the entire ectodomain in a plasma membrane. These simulations were computationally very intensive and did not yield enough data to achieve a satisfactory level of statistical convergence. However, the data we did obtain was consistent with the structural dynamics observed in our fragment-based approach.